



# Evaluating the sensitivity of radical chemistry and ozone formation to ambient VOCs and NOx in Beijing

Lisa K. Whalley[1,2], Eloise J. Slater[1], Robert Woodward-Massey[1,a], Chunxiang Ye[1,a], James D Lee[3,4], Freya Squires[4], James R. Hopkins[3,4], Rachel E Dunmore[4], Marvin Shaw[3,4], Jacqueline F. Hamilton[4], Alastair C Lewis[3,4], Archit Mehra[5,b], Stephen D. Worrall[5,c], Asan Bacak[5,d], Thomas J. Bannan[5], Hugh Coe[5,6], Bin Ouyang[7,e], Roderic L. Jones[7], Leigh R. Crilley[8,f], Louisa J. Kramer[8], William J. Bloss[8], Tuan Vu[8], Simone Kotthaus[9],[10] Sue Grimmond[9] Yele Sun[11], Weiqi Xu[11], Siyao Yue[11], Lujie Ren[11], W. Joe F. Acton[12], C. Nicholas Hewitt[12], Xinming Wang[13], Pingqing Fu[14] and Dwayne E. Heard[1]

[1]School of Chemistry, University of Leeds, Leeds, LS2 9JT, UK

[2]National Centre for Atmospheric Science, University of Leeds, Leeds, LS2 9JT, UK

[3]National Centre for Atmospheric Science, University of York, Heslington, York, YO10 5DD, UK

[4]Wolfson Atmospheric Chemistry Laboratories, Department of Chemistry, University of York, Heslington, York, 10 YO10 5DD, UK

[5]Centre for Atmospheric Science, School of Earth and Environmental Sciences, The University of Manchester, Manchester, M13 9PL, UK

[6]National Centre for Atmospheric Science, University of Manchester, Manchester, M13 9PL, UK

[7]Department of Chemistry, University of Cambridge, UK

[8]School of Geography, Earth and Environmental Sciences, University of Birmingham, B15 2TT, Birmingham, UK

[9]Department of Meteorology, University of Reading, Reading, UK

[10]Institut Pierre Simon Laplace, École Polytechnique, Palaiseau, France

[11]State Key Laboratory of Atmospheric Boundary Layer Physics and Atmospheric Chemistry, Institute for Atmospheric Physics, Chinese Academy of Sciences, 40 Huayanli, Chaoyang District, Beijing 100029, China

[12]Lancaster Environment Centre, Lancaster University, Lancaster, LA1 4YW, UK

[13]State Key Laboratory of Organic Geochemistry, Guangzhou Institute of Geochemistry, Chinese Academy of Sciences, 511 Kehua Street, Wushan, Tianhe District, Guangzhou, GD 510640 , China

[14]Institute of Surface-Earth System Science, Tianjin University, Tianjin 300072, China

[a]Now at: College of Environmental Sciences and Engineering, Peking University, Beijing, 100871, China

[b]Now at: Faculty of Science and Engineering, University of Chester, CH2 4NU, UK

[c]Now at: Aston Institute of Materials Research, School of Engineering and Applied Science, Aston University, Birmingham, B4 7ET, UK

[d]Now at: Turkish Accelerator & Radiation Laboratory, Ankara University Institute of Accelerator Technologies, Atmospheric and Environmemtal Chemistry Laboratory, Gölbaşı Campus, Ankara, Turkey

[e]Now at: Lancaster Environment Centre, Lancaster University, Lancaster, LA1 4YW, UK

[f]Now at: Department of Chemistry, York University, Toronto ON, M3J 1P3, Canada

*Correspondence to:* Lisa Whalley (l.k.whalley@leeds.ac.uk)



**Abstract.** Measurements of OH, HO$_2$, RO$_2$-complex (alkene and aromatic-related RO$_2$) and total RO$_2$ radicals taken during the AIRPRO campaign in central Beijing in the summer of 2017, alongside observations of OH reactivity are presented. The concentrations of radicals were elevated with OH reaching up to 2.8 x 10$^7$ molecule cm$^{-3}$, HO$_2$ peaked at 1 x 10$^9$ molecule cm$^{-3}$ and the total RO$_2$ concentration reached 5.5 x 10$^9$ molecule cm$^{-3}$. OH reactivity (k(OH)) peaked at 89 s$^{-1}$ during the night,

with a minimum during the afternoons of ~22 s$^{-1}$ on average.  An experimental budget analysis, in which the rates of production and destruction of the radicals are compared, highlighted that although the sources and sinks of OH were balanced under high NO concentrations, the OH sinks exceeded the known sources (by 15 ppbv hr$^{-1}$) under the very low NO conditions (<0.5 ppbv) experienced in the afternoons, demonstrating a missing OH source consistent with previous studies under high volatile organic compound (VOC), low NO loadings. Under the highest NO mixing ratios (104 ppbv), the HO$_2$ production rate exceeded the

rate of destruction by ~ 50 ppbv hr$^{-1}$, whilst the rate of destruction of total-RO$_2$ exceeded the production by the same rate indicating that the net propagation rate of RO$_2$ to HO$_2$ may be substantially slower than assumed. If just 10% of the RO$_2$ radicals propagate to HO$_2$ upon reaction with NO, the HO$_2$ and RO$_2$ budgets could be closed at high NO, but at low NO this lower RO$_2$ to HO$_2$ propagation rate revealed a missing RO$_2$ sink that was similar in magnitude to the missing OH source. A detailed box model that incorporated the latest MCM chemical mechanism (MCM3.3.1) reproduced the observed OH

concentrations well, but over-predicted the observed HO$_2$ under low concentrations of NO (<1 ppbv) and under-predicted RO$_2$ (both the complex-RO$_2$ fraction and other RO$_2$ types which we classify as simple-RO$_2$) most significantly at the highest NO concentrations. The model also under-predicted the observed k(OH) consistently by ~10 s$^{-1}$ across all NO$_x$ levels highlighting that the good agreement for OH was fortuitous due to a cancellation of missing OH source and sink terms in its budget. Including heterogeneous loss of HO$_2$ to aerosol surfaces did reduce the modelled HO$_2$ concentrations in-line with the

observations, but only at NO mixing ratios <0.3 ppbv. The inclusion of Cl atoms, formed from the photolysis of nitryl chloride, enhanced the modelled RO$_2$ concentration on several mornings when the Cl atom concentration was calculated to exceed 1 x 10$^4$ atoms cm$^{-3}$ and could reconcile the modelled and measured RO$_2$ concentrations at these times. However, on other mornings, when the Cl atom concentration was lower, large under-predictions in total RO$_2$ remained. Furthermore, the inclusion of Cl atom chemistry did not enhance the modelled RO$_2$ beyond the first few hours after sunrise and so was unable to resolve the

modelled under-prediction in RO$_2$ observed at other times of the day.  Model scenarios, in which missing VOC reactivity was included as an additional reaction that converted OH to RO$_2$, highlighted that the modelled OH, HO$_2$ and RO$_2$ concentrations were sensitive to the choice of RO$_2$ product. The level of modelled to measured agreement for HO$_2$ and RO$_2$ (both complex and simple) could be improved if the missing OH reactivity formed a larger RO$_2$ species that was able to undergo reaction with NO, followed by isomerisation reactions reforming other RO$_2$ species, before eventually generating HO$_2$. In this work an

α-pinene-derived RO$_2$ species was used as an example. In this simulation, consistent with the experimental budget analysis, the model underestimated the observed OH indicating a missing OH source. The model uncertainty, with regards to the types of RO$_2$ species present and the radicals they form upon reaction with NO (HO$_2$ directly or another RO$_2$ species), leads to over an order of magnitude less O$_3$ production calculated from the predicted peroxy radicals than calculated from the observed





peroxy radicals at the highest NO concentrations. This demonstrates the rate at which the larger $RO_2$ species propagate to $HO_2$
or to another $RO_2$ or indeed to OH needs to be understood to accurately simulate the rate of ozone production in environments
such as Beijing where large multifunctional VOCs are likely present.

# 1 Introduction

Owing to strict emission controls being implemented across China, a reduction in the levels of $PM_{10}$, $PM_{2.5}$ and $SO_2$ have been
observed in the country since 2013 (Huang et al., 2018). Similar reductions in these primary pollutants are echoed in other
countries across the globe. In the US this reduction in primary emissions is reflected in a ~0.4 ppbv $yr^{-1}$ reduction in peak $O_3$
(He et al., 2020). In China, however, despite reductions in primary emissions, the concentration of ground-level ozone has
been gradually increasing between 2013 – 2017 (Huang et al., 2018). The highest peak ozone concentrations in China are
observed in the Beijing area (Wang et al., 2017a) where the highest $O_3$ mixing ratio of 286 ppbv was recorded at a rural site
50 km north of the centre (Wang et al., 2006). During the Beijing Olympic Games, despite emission controls, hourly ozone
mixing ratios between 160 to 180 ppbv were frequently observed in central Beijing (Wang et al., 2010). Ozone is a secondary
pollutant, primarily formed in the troposphere via OH-initiated VOC oxidation in the presence of $NO_x$. $O_3$ concentrations in
megacities worldwide frequently exceed regulatory limits during the summer months, with elevated ozone concentrations
shown to have negative impacts on human and crop health. The radical species, OH, $HO_2$ and $RO_2$ play a central role in the
catalytic photochemical cycle which removes primary emissions and leads to ozone formation. The OH radical initiates the
oxidation of VOCs leading to the formation of peroxy radicals ($HO_2$ and $RO_2$). Peroxy radicals oxidise NO to $NO_2$ which
photolyses and generates ozone. Under high $NO_x$ conditions, OH preferentially reacts with $NO_2$, and both peroxy radical
production (via VOC oxidation) and, in turn, ozone production decreases. This non-linear relationship between ozone and $NO_x$
complicates efforts to reduce the ambient ozone levels as, in $NO_x$-saturated environments, reductions in $NO_x$ can lead to
increases in the rate of ozone production (e.g. (Bigi and Harrison, 2010)). Furthermore, a number of studies have highlighted
that efforts to reduce PM have the potential to exacerbate $O_3$ due to concomitant increases in $HO_2$ caused by a reduction in the
heterogeneous loss of $HO_2$ to aerosol surfaces (Li et al., 2019), although there is continued debate on the magnitude of this
effect from field studies (Tan et al., 2020). As well as the central role OH plays in photochemical ozone formation, OH
promotes the formation of secondary aerosols (sulphate, nitrate and secondary organic aerosols (SOA)) which have negative
impacts on human health (Chen et al., 2013). Large, complex $RO_2$ radicals are precursors to highly oxidised molecules (HOMs)
(Ehn et al., 2014) also which have been shown to condense and contribute to SOA (Mohr et al., 2019). In China, the fraction
of PM attributed to secondary aerosols is significant (between 44 – 71%, (Huang et al., 2014)) and so understanding the
oxidation chemistry which converts primary emissions to secondary aerosols is an ongoing challenge. There has been an
increasing growth in photochemical oxidant studies conducted in China where radical observations have been performed over
the past decade with the PKU and Juelich groups leading these efforts. The first radical observations took place in the summer





of 2006 with observations made in the Pearl River Delta region (Hofzumahaus et al., 2009; Lu et al., 2012) (PRIDE-PRD-2006) and also in suburban Beijing (Lu et al., 2013) (CareBeijing2006). These campaigns revealed a strong atmospheric oxidation capacity with elevated levels of OH and $HO_2$ in these regions, with OH concentrations up to $2.6 \times 10^7$ molecule $cm^{-3}$ and $HO_2$ concentrations up to $2.5 \times 10^9$ molecule $cm^{-3}$ reported (Lu et al., 2012). Even during the wintertime, under low

levels of solar radiation, concentrations of OH can reach $3 \times 10^6$ molecule $cm^{-3}$ in Beijing (Slater et al., 2020) which is similar to the OH concentrations observed in other urban centres in European cities during the summer months (Whalley et al., 2018). Similar to findings from radical observations and subsequent modelling activities in forested regions (Whalley et al., 2011), which are characterised by high VOC emissions and relatively low $NO_x$ concentrations, the observations and modelling studies in China in summer (Hofzumahaus et al., 2009; Lu et al., 2012; Lu et al., 2013) revealed that the high OH concentrations could

only be explained if an additional source of OH, from recycling peroxy radicals to OH, was added to the model. An updated isoprene scheme (Peeters et al., 2009; Peeters et al., 2014) which included isomerisation reactions of the isoprene-derived $RO_2$ radicals, was unable to reconcile the OH observations, however. In a subsequent field study conducted in the PRD region (Tan et al., 2019), $RO_2$ observations were made using the $RO_x$LIF technique alongside OH, $HO_2$ and OH reactivity allowing an experimental budget analysis for OH, $HO_2$, $RO_2$ and $RO_x$ (OH + $HO_2$ + $RO_2$) to be performed. The analysis demonstrated a

missing OH source of $4 - 6$ ppbv $hr^{-1}$ and a missing $RO_2$ sink that was similar in magnitude and, hence, supports the hypothesis of a missing mechanism that converts $RO_2$ species to OH under low NO conditions. The authors calculated that the unknown $RO_2$ to OH conversion that does not involve reaction with NO (and, therefore, does not lead to the formation of ozone) reduced ozone production by 30 ppbv per day demonstrating that knowledge of the branching ratio between the competitive reactions that $RO_2$ radicals undergo (bimolecular reaction with NO or unimolecular isomerisation), as well as the overall VOC oxidation

rate, is important when determining in situ ozone production.

In a recent campaign conducted at a rural site in the North China Plain (Tan et al., 2017), during periods for which NO mixing ratios were below 300 pptv, an additional OH recycling mechanism was again needed to reconcile the OH concentrations observed. The modelled $RO_2$ concentrations were in good agreement with those observed under low NO concentrations typically experienced during the afternoon, however, the model under-predicted the $RO_2$ concentrations by a factor of 3 - 5 at

the higher NO mixing ratios (>1 ppbv) that were observed during the mornings. Additional sources of $RO_2$ from the photolysis of $ClNO_2$ and subsequent reactions of Cl atoms with VOCs, as well as $RO_2$ from the missing reactivity determined, could explain ~ $10 - 20\%$ of the model under-prediction, but could not fully resolve the missing $RO_2$ source of 2ppbv $hr^{-1}$ under the high NO conditions. As a result, the model was found to under-predict the net in situ chemical ozone production by 20 ppbv per day. In London, during the ClearfLo campaign (Whalley et al., 2018), under higher NO mixing ratios (> 3ppbv) a box

model constrained to the MCM3.2 was found to increasingly under-predict the $RO_2$ concentrations observed with $NO_x$ and, as a consequence the rate of ozone production calculated from the modelled peroxy radical concentrations was up to an order of magnitude lower than the ozone production rate calculated from the observed peroxy radicals. The model was able to reproduce



the observed levels of HO$_2$ under the high NO concentrations, but over-predicted HO$_2$ concentrations when NO mixing ratios were below 1 ppbv and modest under-predictions of OH were observed under low NO conditions which demonstrated uncertainties in radical cycling at low NO. Conversely, in other urban studies, models were found to increasingly under-predict HO$_2$ as NO$_x$ levels increased beyond ~ 1 ppbv (Martinez et al., 2003; Ren et al., 2013; Brune et al., 2016) although in some of these earlier studies, the HO$_2$ observations may have been influenced by an RO$_2$ interference (Whalley et al., 2013). Understanding the cause of the model failure under different NO regimes in urban centres is critical to be able to accurately predict ozone production and to determine ozone abatement strategies that can be implemented to successfully reduce ozone levels. Measurements of OH, HO$_2$ and RO$_2$ as well as OH reactivity are necessary to fully explore a model's skill to capture the entire atmospheric oxidation cycle and to begin to identify mechanisms that can reconcile the concentration of all radical species.

The integrated Study of AIR Pollution PROcesses in Beijing (AIRPRO) project involved two intensive measurement periods that took place in central Beijing during the winter of 2016 and during the following summer of 2017 and was part of the larger Air Pollution and Human Health (APHH) program. APHH had the overall aim of better understanding the sources, atmospheric transformations and health impacts of air pollutants in Beijing to improve air quality forecasting capabilities (Shi et al., 2019). In this paper the observations of OH, HO$_2$, RO$_2$ and OH reactivity from the summer period are compared to a detailed zero dimensional box model run with the latest Master Chemical Mechanism (MCM3.3.1) and an experimental budget analysis is performed on all radical species. The overall objective of this research was to test the model's ability to reproduce the radical concentrations and, through the budget analysis investigate the balance between radical production and destruction rates. Following on from the results of earlier radical observation and modelling studies conducted in urban regions, this research will investigate if there are missing radical sources and sinks under different NO regimes and investigate new chemistry that may improve model predictions. We will assess how uncertainties in the model mechanism influence the rate of in situ ozone production in an environment with large and complex VOC emissions and under highly variable NO$_x$ concentrations.

## 2 Experimental

### 2.1 Site description

The observations took place in central Beijing at the Institute of Atmospheric Physics (IAP), which is part of the Chinese Academy of Sciences. The site was located between the third and fourth north ring roads in Beijing and was within 150 m of several busy roads. All instrumentation was located in close proximity within 9 shipping containers that were placed on a





grassed area surrounding a large (325 m) meteorological tower. Further details of the measurement site and an overview of all
the instrumentation that was run during the campaign can be found in Shi et al., (2019).

## 2.2 FAGE instrumentation

The University of Leeds fluorescence assay by gas expansion (FAGE) instrument was deployed at the IAP site and made
measurements of OH, $HO_2$, $RO_2$ radicals and OH reactivity (k(OH)). The instrumental set-up was analogous to that used during

the ClearfLo project (see Whalley et al., (2016) for the k(OH) instrument description and Whalley et al., (2018) for the OH,
$HO_2$, and $RO_2$ instrument details) and also the winter AIRPRO project (Slater et al., 2020) and so is only briefly overviewed
here. Two detection cells, the $HO_x$ cell and the $RO_x$LIF cell, were located on the roof of the Leeds FAGE shipping container
at a sampling height of 3.5 m. The k(OH) instrument, which was housed inside the container, alongside all other FAGE
instrument components (including the laser system), drew air from close by the radical detection cells via an ½" Teflon line.

The $HO_x$ cell made sequential measurements of OH and then the sum of OH + $HO_2$, by the addition of NO (Messer, 99.95%)
which titrated $HO_2$ to OH for detection by laser induced fluorescence (LIF). In the $RO_x$LIF reactor, in $HO_x$-mode, a flow of
CO (10% in $N_2$) was added just beneath the sampling inlet and this rapidly converted any ambient OH sampled to $HO_2$. Within
the $RO_x$LIF FAGE cell, a continuous flow of NO (99.95%) titrated ambient $HO_2$, the converted OH and also a large % of $RO_2$-
complex radicals (see below) to OH for detection. In $RO_x$-mode, a total-$RO_2$ + $HO_2$ + OH measurement was made by addition

of a dilute flow of NO (500 ppmv in $N_2$) alongside the CO which promoted the conversion of all $HO_2$ and $RO_2$ radicals to OH;
the OH formed was rapidly re-converted to $HO_2$ by reaction with CO. Within the $RO_x$LIF FAGE cell, the $HO_2$ was titrated
back to OH, by reaction with NO, for detection. Using the methodology outlined in Whalley et al., (2013) the sensitivity of
both the $HO_x$ and $RO_x$LIF FAGE cells towards $HO_2$ and $RO_2$-complex species was assessed before the instrument was
deployed to Beijing by sampling isoprene-derived $RO_2$; the sensitivity of the $HO_x$ cell towards other $RO_2$ types such as those

derived from ethene, methanol and propane has been previously conducted (Whalley et al., 2013) and compared well with
model-predicted sensitivities. The sensitivity of the ROxLIF instrument has also been assessed previously towards a range of
$RO_2$ types deriving from methane, isoprene, ethene, toluene, butane and cyclohexane and, again, compared well with model-
predicted sensitivities (Whalley et al., 2018). $RO_2$-complex refers to any $RO_2$ species (primarily those derived from alkene and
aromatic hydrocarbons) that have the potential to decompose into OH in the presence of NO on the time-scale of the FAGE

residence time and, therefore, have the potential to act as an $HO_2$ interference. The NO flow in the $HO_x$ cell was kept low to
minimise the conversion efficiency of $RO_2$-complex to OH and the conversion efficiency was found to be <5% when isoprene-
derived $RO_2$ radicals were sampled. In the $RO_x$LIF FAGE cell, a higher NO flow was employed to promote the conversion of
$RO_2$-complex to OH, enabling 89% of isoprene-derived $RO_2$ radicals to be detected. From the relative sensitivities of the two



cells to OH, HO$_2$ and RO$_2$-complex, and by subtraction of RO$_2$-complex from total RO$_2$, the concentration of RO$_2$ species that

do not act as an HO$_2$ interference (RO$_2$-simple) has been derived.

For the entirety of the campaign, the HO$_x$ cell was equipped with an inlet-pre-injector (IPI) (Woodward-Massey et al., 2020) which, by injection of propane into the ambient air-stream directly above the HO$_x$ inlet, removes ambient OH and enables a background measurement from laser scatter, solar scatter and detector dark counts (and potentially any cell-generated OH) to be determined whilst the laser is tuned to the OH transition. The subtraction of this background signal from the ambient OH

signal provides the OH$_{CHEM}$ measurement which can be compared to the traditional OH$_{WAVE}$ measurement in which the background signal (from laser scatter, solar scatter and detector dark counts only) is determined by tuning the laser wavelength away from the OH transition. Differences between OH$_{CHEM}$ and OH$_{WAVE}$ can highlight the presence of an OH interference. During the summer AIRPRO campaign, once the known OH interference deriving from laser-photolysis of ambient ozone and the subsequent reaction of photogenerated O($^1$D) atoms with ambient H$_2$O (v) was accounted for (Woodward-Massey et al.,

2020) the agreement between OH$_{CHEM}$ and OH$_{WAVE}$ was generally very good (see figure 14 in Woodward-Massey et al., (2020)). However, on five afternoons when ozone was extremely elevated (>100 ppbv) and OH concentrations were high (>1x10$^7$ cm$^{-3}$), OH$_{WAVE}$ was greater than OH$_{CHEM}$ (by up to 18 %) highlighting a small unknown interference under these very perturbed conditions. In all the model-measurement comparisons presented in the section 3, the interference-free OH$_{CHEM}$ measurement is used.

Both detection cells were calibrated every 3 days during the campaign by photolysis of a known concentration of H$_2$O (v) at 185 nm with a Hg lamp in synthetic air (Messer, Air Grade Zero 2) within a turbulent flow tube which generates an equal concentration of OH and HO$_2$ (Whalley et al. 2018). The product of the photon flux at 185 nm (determined by N$_2$O actinometry (Commane et al., 2010) before and after the instrument was deployed to Beijing), [H$_2$O] and irradiance time, was used to calculate [OH] and [HO$_2$]. For calibration of RO$_2$ concentrations, methane (Messer, Grade 5, 99.99%) was added to the

humidified air flow in sufficient quantity to completely convert OH to CH$_3$O$_2$. The median limit of detection (LOD) during the campaign was $6.1 \times 10^5$ molecule cm$^{-3}$ for OH, $2.8 \times 10^6$ molecule cm$^{-3}$ for HO$_2$ and $7.2 \times 10^6$ molecule cm$^{-3}$ for CH$_3$O$_2$ at a typical laser power of 11 mW for a 5 minute data acquisition cycle (SNR=2). The field measurements of all species were recorded with 1 s time-resolution, and the precision of the measurements was calculated using the standard errors in both the online and offline points. The accuracy of the measurements was ~ 26 % (2σ), and is derived from untertainties in the

calibration, which derives largely from that of the chemical actinometer (Commane et al., 2010).

**2.3 Experimental budget analysis**

An experimental budget analysis has been conducted for OH, HO$_2$, RO$_2$, and total RO$_x$ following the approach outlined in Tan et al., (2019) and which relies only on field-measured quantities (concentrations and photolysis rates) and published chemical





kinetic data, and not any model calculated concentrations. The rates of production and destruction of each radical species is
calculated using equations 1 – 8 below.

$$P_{\text{OH}} = j_{\text{HONO}}[\text{HONO}] + (2f \times j_{\text{O}^1\text{D}}[\text{O}_3]) + \sum i\{\varphi^i{}_{\text{OH}}k^i{}_1[\text{alkene}]_i[\text{O}_3]\} + (k_2[\text{NO}] + k_3[\text{O}_3])[\text{HO}_2] \tag{1}$$

$$D_{\text{OH}} = [\text{OH}]k_{\text{OH}} \tag{2}$$

$$P_{\text{HO}_2} = 2j_{\text{HCHO\_r}}[\text{HCHO}] + \sum i\left\{\varphi^i{}_{\text{HO}_2}k^i{}_1[\text{alkene}]_i[\text{O}_3]\right\} + (k_4[\text{HCHO}] + k_5[CO])[OH] + \alpha k_6[\text{NO}][\text{RO}_2] \tag{3}$$

$$D_{\text{HO}_2} = (k_7[\text{NO}] + k_8[\text{O}_3] + k_9[\text{RO}_2] + k_{\text{het}} + 2k_{10}[\text{HO}_2])[\text{HO}_2] \tag{4}$$

$$P_{\text{RO}_2} = \sum i\left\{\varphi^i{}_{\text{RO}_2}k^i{}_1[\text{alkene}]_i[\text{O}_3]\right\} + k_{\text{OH}}[\text{VOC}][\text{OH}] \tag{5}$$

$$D_{\text{RO}_2} = \{(\alpha + \beta)k_6[\text{NO}] + (2k_{11}[\text{RO}_2] + k_9[\text{HO}_2])[\text{RO}_2]\} \tag{6}$$


$$P_{\text{RO}_x} = j_{\text{HONO}}[\text{HONO}] + 2f \times j_{\text{O}^1\text{D}}[\text{O}_3] + 2j_{\text{HCHO\_r}}[\text{HCHO}] + \sum i\left\{(\varphi^i{}_{\text{OH}} + \varphi^i{}_{\text{HO}_2} + \varphi^i{}_{\text{RO}_2})k^i{}_1[\text{alkene}]_i[\text{O}_3]\right\} \tag{7}$$

$$D_{\text{RO}_x} = (k_{12}[\text{NO}_2] + k_{13}[\text{NO}])[\text{OH}] + \beta k_6[\text{NO}][\text{RO}_2] + 2(k_{11}[\text{RO}_2]^2 + k_9[\text{RO}_2][\text{HO}_2] + k_{10}[\text{HO}_2]^2) \tag{8}$$

where $j_{\text{HONO}}$ and $j_{\text{O}^1\text{D}}$ are the measured photolysis rates of HONO and $\text{O}_3$ (forming $\text{O}^1\text{D}$) respectively, $f$ is the fraction of $\text{O}^1\text{D}$
radicals that react with $\text{H}_2\text{O}$ rather than are collisionally quenched to $\text{O}(^3\text{P})$ ($f = 0.1$ on average) and $\varphi^i{}_{\text{OH}}$, $\varphi^i{}_{\text{HO}_2}$, $\varphi^i{}_{\text{RO}_2}$ and
$k^i{}_1$ are the yield of OH, $\text{HO}_2$ and $\text{RO}_2$ from, and rate coefficients for, individual ozone-alkene reactions taken from the
MCM3.3.1 respectively. $j_{\text{HCHO\_r}}$ is the measured HCHO photolysis rate that yields $\text{HO}_2$ radicals, $k_{\text{het}}$ is the first order loss of
$\text{HO}_2$ to the measured aerosol surface area, calculated using Eq.9:

$$k_{\text{het}} = \frac{\omega A \gamma}{4} \tag{9}$$

where $\omega$ is the mean molecular speed of $\text{HO}_2$ (equal to 43725 cm s$^{-1}$ at 298 K), $\gamma$ is the aerosol uptake coefficient (0.2 is used
here as recommended by Jacob (2000)) and $A$ is the measured aerosol surface area in cm$^2$cm$^{-3}$. $\alpha$ is the fraction of $\text{RO}_2$ radicals
that upon reaction with NO propagate to $\text{HO}_2$ rather than reform another $\text{RO}_2$ radical; initially $\alpha = 1$ has been assumed. $\beta$ is
the fraction of $\text{RO}_2$ radicals that upon reaction with NO form alkyl nitrates and is set to 0.05 as used by Tan et al., (2019) to
represent an average alkyl nitrate yield for the various types of $\text{RO}_2$ species likely present. All rate coefficients ($k_1 - k_{13}$) used
are listed in Table 1 and the concentration of species used in the budget analysis are the concentrations that were observed
during the campaign.



## 2.4 MCM3.3.1 box model description

A zero-dimensional (box) model incorporating the Master Chemical Mechanism (MCM3.3.1) (Jenkin et al., 2015) (http://mcm.leeds.ac.uk/MCM/home) was used to predict the radical concentrations and OH reactivity for comparison with the observations. The model was constrained by measurements of NO, $NO_2$, $NO_3$, $O_3$, CO, HCHO, $HNO_3$, HONO, water vapour, temperature, pressure and individual VOC species measured by DC-GC-FID (dual-channel gas chromatography with flame ionisation) and PTR-ToF-MS (proton transfer reactor-time of flight-mass spectrometry). Table 2 lists the different VOC species measured. HCHO was measured using a recently developed LIF instrument with 1 sec time resolution and LOD of 80

pptv (Cryer, 2016). HONO was measured by a long-path absorption photometer (LOPAP) and broadband cavity-enhanced absorption spectrophotometry (BBCEAS) and the HONO concentration as recommended in Crilley et al., (2019) are used here. Further details on all instrumentation deployed during the campaign is overviewed in Shi et al. (2019).

The model was constrained with the measured photolysis frequencies $j(O^1D)$, $j(NO_2)$ and $j(HONO)$), which were calculated from the measured wavelength-resolved actinic flux and published absorption cross sections and photodissociation quantum

yields. For other species which photolyse at near-UV wavelengths (<360 nm), such as HCHO and $CH_3CHO$, the photolysis rates were calculated by scaling to the ratio of clear-sky $j(O^1D)$ to observed $j(O^1D)$ to account for clouds. For species which photolyse further into the visible the ratio of clear-sky $j(NO_2)$ to observed $j(NO_2)$ was used. The variation of the clear-sky photolysis rates (j) with solar zenith angle ($\chi$) was calculated within the model using the following expression:

$$j = l \cos(\chi)m \times e^{-n \sec(\chi)} \tag{10}$$

with the parameters $l$, $m$ and $n$ optimised for each photolysis frequency (see Table 2 in Saunders et al., (2003)). The model inputs were updated every 15 minutes, the species that were measured more frequently were averaged to 15 minutes whilst the measurements with lower time resolution were interpolated. To estimate how long model generated intermediate species survive before being physically removed by processes such as deposition or ventilation, the model was left unconstrained to glyoxal and the rate of physical loss was varied. The model was able to reproduce the observed glyoxal concentrations if a

deposition velocity of 0.5 cm $s^{-1}$ was used, combined with a ventilation term that increased with the measured boundary layer depth (Kotthaus and Grimmond, 2018). As the boundary layer gradually increased in the morning, the lifetime of glyoxal with respect to ventilation was ~ 1 hour, whilst at night the lifetime gradually increased to ~ 5 hours; this variable lifetime was applied to all model-generated species. As a further check on the physical loss rate imposed, the model was run unconstrained to HCHO using the same deposition rates and was found to reproduce the observed HCHO concentrations that were observed

during the daytime, but under-predicted the concentrations at night, potentially indicating that primary emissions of HCHO as



well as secondary production contributed to the observed concentrations. In all the model scenarios presented in section 3, the observed HCHO concentration is used. The model was run for the entirety of the campaign in overlapping 7 day segments. To allow all the unmeasured, model generated intermediate species time to reach steady state concentrations, the model was initialised with inputs from the first measurement day and spun-up for 2 days before comparison to measurements of OH, HO$_2$,

RO$_2$ and k(OH) were made. For comparison of the modelled RO$_2$ to the observed RO$_2$-total, RO$_2$-complex and RO$_2$-simple, the RO$_x$LIF instrument sensitivity towards each RO$_2$ species in the model was determined by running a model first under the RO$_x$LIF reactor and then the RO$_x$LIF FAGE cell conditions (NO concentrations and residence times) to determine the conversion efficiency of each modelled RO$_2$ species to HO$_2$.

### 2.4.1 Model descriptions

A series of model runs have been performed and are summarised in Table 3:

### 3 Results and Discussion

### 3.1 Overview of the chemistry and meteorology during the campaign

As part of the AIRPRO project, gas-phase, aerosol, and meteorological observations were made at the IAP site from the 21$^{st}$ May to 26$^{th}$ June in 2017. Typically clear skies and elevated temperatures prevailed, with rain on just a few days. Temperatures

frequently exceeded 35 °C whilst j(O$^1$D) peaked at just over 3 x 10$^{-5}$ s$^{-1}$ at noon (Figure 1). The dominant wind direction reaching the site during the summer was from the southwest and the measured hourly mean wind speed was 3.6 ms$^{-1}$ (Shi et al., 2019). Despite the close proximity of the measurement site to the heavily trafficked Jingzang highway in Beijing, mixing ratios of NO, which were elevated during the morning hours, often dropped below 500 pptv during the afternoon. The daytime emissions of NO$_x$ that were recorded during the project displayed a rapid increase at 05:00 and then remained reasonably

constant throughout the day, with a mean flux value of 4.6 mg m$^{-2}$ hr$^{-1}$, before dropping again at 17:00 (Squires et al., 2020). The rapid decrease in NO into the afternoon, therefore, was not driven by a change in emissions, but rather instead by the increasing boundary layer depth and also by the chemistry, as elevated levels of ozone observed in the afternoon effectively titrated NO to NO$_2$ (Newland et al., 2020). Isoprene mixing ratios also peaked in the afternoon, often reaching a few ppbv, indicative of a biogenic source. The variation in NO$_x$ and VOC concentrations experienced at the site provides an opportunity

to assess the skill of the MCM to capture the complex chemistry occurring over an extremely wide range of chemical regimes that encompasses both typical urban conditions (high NO$_x$) as well as chemical conditions more akin to forested environments (low NO, high BVOC). From the 9$^{th}$ – 12$^{th}$ June, NO levels were elevated throughout the day suggesting a local source, whilst from the 17$^{th}$ June to the end of the measurement period, NO concentrations dropped and, so as well as the strong diurnal trend observed in the NO concentration, these periods provide further opportunity to test the model's ability to predict radical





concentrations as a function of NO by removing concomitant variables such as changing boundary layer depth and sunrise which occurred in unison with the morning increase in NO concentration.

## 3.2 Radical concentrations and OH reactivity

The concentrations of $RO_x$ ($OH + HO_2 + RO_2$) radicals were high during the campaign (Figure 2), with OH concentrations
frequently exceeded $1 \times 10^7$ molecule $cm^{-3}$ and reaching up to $2.8 \times 10^7$ molecule $cm^{-3}$ on the 30[th] May. These OH levels are amongst the highest measured in an urban environment (Lu et al., 2019), and are comparable to the OH concentrations observed in the Pearl River Delta downwind of the Southern Chinese megacity of Guangzhou, where OH concentrations reached $2.6 \times 10^7$ molecule $cm^{-3}$ (Lu et al., 2012). $HO_2$ concentrations peaked at $1 \times 10^9$ molecule $cm^{-3}$ on the 9[th] June, whist the highest concentrations of total $RO_2$ were observed during the latter half of the campaign, peaking at $5.5 \times 10^9$ molecule $cm^{-3}$ on the
afternoon of the 15[th] June. $RO_2$ measurements, alongside OH and $HO_2$, were, until recently, relatively rare. OH and $RO_x$ were measured during the MEGAPOLI project in Paris (Michoud et al., 2012) where the average daytime maximum concentrations of $RO_x$ were $1.2 \times 10^8$ molecule $cm^{-3}$ which is over an order of magnitude lower than the levels observed in Beijing. Since the development of the $RO_x$LIF technique, (Fuchs et al., 2008), $RO_2$ observations are now reported by the Leeds, Juelich and PKU FAGE groups. $RO_2$ concentrations observed in London in the summer reached up to $5.5 \times 10^8$ molecule $cm^{-3}$ in air masses that
had previously passed over central London (Whalley et al., 2018). In Wangdu, a town situated on the North China Plain, 170 km northeast of Beijing, summertime $RO_2$ concentrations reached up to $1.5 \times 10^9$ molecule $cm^{-3}$ (Tan et al., 2017) which, although lower than observed in central Beijing, are much higher than observed in the summertime in European cities suggesting that there may be significant differences in the urban photochemistry occurring in China and Europe.

As well as the elevated daytime radical concentrations, concentrations of OH, $HO_2$ and $RO_2$ remained elevated above the instrumental LOD on most nights. The high night-time OH concentrations (ranging from the LOD up to $2 \times 10^6$ molecule $cm^{-3}$) are comparable to the levels of OH observed at night in Yufa (a suburb of Beijing) and downwind of Guangzhou where night-time OH concentrations ranged from $0.5 – 3 \times 10^6$ molecule $cm^{-3}$ (Lu et al., 2014). The observations of OH from the earlier China campaigns could be reconciled by a model if an additional $RO_x$ production process was included which recycled
$RO_2$ to OH via $HO_2$. A weak positive correlation is observed between night-time OH and $RO_2$ at night during AIRPRO and the secondary peak in $RO_2$ occurred when $NO_3$ was observed to increase rapidly at ~19:30 suggesting that nitrate chemistry was one source of radicals in the evening. Alkyl nitrates, formed from aldehydes $+ NO_3$ were also enhanced at these times at this site (Reeves et al., 2019).

The OH reactivity, typical of urban environments displayed an inverse relationship with boundary layer height and was highest during the nights when emissions were compressed into a lower boundary layer depth of ~150 m. An average maximum of k(OH) ~37 $s^{-1}$ was observed at 06:00 with OH reactivity reaching 89 $s^{-1}$ on the 15[th] June at 03:00. During the daytime the OH



reactivity dropped to a minimum of ~22 s$^{-1}$ on average at ~15:00 when the boundary layer had increased to ~1500 m. The magnitude of OH reactivity observed during AIRPRO is comparable to the OH reactivity observed at other urban sites in China

in the summer (Lou et al., 2010; Fuchs et al., 2017) and also in Tokyo during the summer (Sadanaga et al., 2004; Chatani et al., 2009). In London, OH reactivity was approximately ~7 – 10 s$^{-1}$ lower than in central Beijing with ~15 s$^{-1}$ observed during the day on average and an average maximum of ~27 s$^{-1}$ at 06:00 (Whalley et al., 2016). Lower OH reactivities are also reported from US urban sites in New York and Texas (Ren et al., 2003; Mao et al., 2010).

**3.3 Experimental Radical Budget Analysis**

Owing to the relatively short-lifetime of radicals, it can be assumed that their production rates and destruction rates are balanced. A comparison of the rates of production and destruction for each radical species can be used to help identify if all radical sources and sinks are accounted for and if the rates of propagation between radical species is fully understood. In

London, the ratio of the OH production rate (Eq. 1) to OH destruction rate (Eq.2) was generally close to 1 throughout the campaign demonstrating consistency between the OH, HO$_2$, k(OH), HONO and NO observations (Whalley et al., 2018). However, under low NO conditions (<0.5 ppbv) the rate of OH destruction exceeded the calculated production rate indicating that Eq.1 was missing a source term under these regimes (Whalley et al., 2018). A steady-state analysis of HO$_2$ conducted for the London project which balanced the HO$_2$ production terms (Eq.3) with the first and second order loss terms (Eq.4)

highlighted that closure between the production an destruction terms could only be reconciled if the rate of propagation of the observed RO$_2$ radicals to HO$_2$ was decreased substantially to just 15%, demonstrating that the mechanism by which RO$_2$ radicals propagate to other radical species may not be well understood (Whalley et al., 2018). As set out by Tan et al., (2019), analogous budget analyses can be performed for RO$_2$ species (Eq.5 – Eq.6) and for the entire RO$_x$ budget (Eq.7 – Eq.8). Tan et al., (2019) found that the production and destruction terms for RO$_2$ were balanced in the mornings in the PRD, when the

measured OH reactivity was used to calculate the rate of RO$_2$ production from VOC+OH reactions, but during the afternoon a missing RO$_2$ sink (2 – 5 ppbv hr$^{-1}$) was evident. In the PRD study (Tan et al., 2019), the OH destruction rate exceeded the production rate by 4 – 6 ppbv hr$^{-1}$ in the afternoon, but, in contrast to London (Whalley et al., 2018), the HO$_2$ budget was closed throughout the whole day. The total rate of RO$_x$ production and destruction were in good agreement in the PRD (Tan et al., 2019).


A comparison of the campaign median production and destruction rates for RO$_x$, OH, HO$_2$ and RO$_2$ during AIRPRO are presented in figure 3. The total rate of RO$_x$ production and destruction are in good agreement throughout the day from ~10 am. A night-time source of radicals of just under 1 ppbv hr$^{-1}$ is missing from the budget analysis likely reflecting missing production from NO$_3$ +VOC reactions (night-time radical production is considered further in section 3.5). From 6 am to 10 am, the RO$_x$

destruction exceeds the production by up to 4 ppbv hr$^{-1}$ indicating a substantial, ~50%, missing primary RO$_x$ source at this time. Previous work has suggested that Cl-initiated VOC oxidation may be an important source of RO$_2$ radicals in urban





regions (Riedel et al., 2014; Bannan et al., 2015; Tan et al., 2017) but has not been included in the $RO_x$ or $RO_2$ production rate calculations here. Nitryl chloride was measured for part of the AIRPRO campaign and the impact of this on the modelled $RO_2$ concentration is investigated in section 3.4. The total $RO_x$ production and destruction rate is of the order of 6 ppbv hr$^{-1}$ at noon

which is slightly faster than in the PRD, where a median peak total radical production rate of ~4 ppbv hr$^{-1}$ was calculated. The median OH destruction rate is ~ 30 ppbv hr$^{-1}$ at noon and is roughly twice as fast the production rate at this time highlighting a large missing source of OH radicals in the budget (~ 15 ppbv hr$^{-1}$). Although a missing OH source was also reported in the PRD (Tan et al., 2019), the missing production rate is ~ 3 times faster during AIRPRO. The known OH production rate during AIRPRO is dominated by the reaction of $HO_2$ with NO (contributing ~60 % during the day to P(OH) in Eq.1). The median

peak $HO_2$ production of ~60 ppbv hr$^{-1}$ is observed in the morning hours and greatly exceeded the known rate of $HO_2$ destruction by ~ 50 ppbv hr$^{-1}$. $HO_2$ production is driven by the reaction of $RO_2$ with NO which accounts for 88% of the total. The reaction of OH with CO and HCHO accounts for a further 9%. The total $HO_2$ production rate is approximately 4 times faster than that calculated for the PRD (Tan et al., 2019). The total rate of $RO_2$ destruction mirrors the $HO_2$ production in that it is dominated by the reaction of $RO_2$ radicals with NO. From sunrise – 14:00 the rate of $RO_2$ destruction is faster than $RO_2$ production by up

to 50 ppbv hr$^{-1}$. After 14:00 the rate of $RO_2$ production and destruction are in good agreement. This trend contrasts with the budget analysis presented from PRD (Tan et al., 2019), which highlighted a possible missing $RO_2$ sink during the afternoon hours and budget closure in the morning hours.

Binning the ratio of P(OH) to D(OH), P($HO_2$) to D($HO_2$)  and P($RO_2$) to D($RO_2$) against NO mixing ratio (figure 4) reveals

that the $RO_2$ budget is in good agreement at the lowest NO mixing ratios but as NO mixing ratios increase the destruction of $RO_2$ becomes faster than production of $RO_2$ by up to a factor of 10 at the highest NO bin. The trends in the $RO_2$ and $HO_2$ ratios are similar in the morning hours, albeit in opposite directions, and suggests that rather than there being a missing primary source of $RO_2$ and missing sink for $HO_2$ that happen to balance, instead, as found in London (Whalley et al., 2018), the net propagation rate of $RO_2$ to $HO_2$ may be substantially slower than the rate that has currently been used in this analysis. In

London (Whalley et al., 2018), the modelled rate of production analysis revealed that only ~50% of the total $RO_2$ species propagated to $HO_2$ following reaction with NO, as a significant fraction of the alkoxy radicals formed (such as those generated during the oxidation of monoterpenes and long-chain alkanes) preferentially isomerised and reformed a more oxidised $RO_2$ species in the presence of $O_2$ instead. In the London model radical flux analysis using the MCM3.2 , (Whalley et al., 2018), the propagation of alkyl- and acyl-$RO_2$ species were combined and so the interconversion of acyl-$RO_2$ radicals (from the OH-

initiated oxidation of aldehydic VOCs, photolysis of ketones and decomposition of PAN species) to alkyl-$RO_2$ radicals following reaction with NO was not explicitly shown, but this interconversion of one $RO_2$ species to another would serve to reduce the fraction of $RO_2$ radicals that propagate to $HO_2$ further. Thus far for AIRPRO, the experimental budget analysis has assumed that 95% of the measured $RO_2$ species, upon reaction with NO, produce $HO_2$. If, however, a large fraction of the total $RO_2$ measured derive from long-chain alkanes, monoterpenes or acyl-$RO_2$ species, the budget analysis will over-estimate $HO_2$



production and also the net $RO_2$ destruction, as the reaction of these peroxy radicals with NO effectively converts one $RO_2$ species to another $RO_2$ species, and so the reaction with NO will be neutral in terms of $RO_2$ production and destruction. Taking $\alpha = 0.1$ leads to a good agreement between the production and destruction rates of $HO_2$ over the whole day and the observed range of NO. The production and destruction rates of $RO_2$ agree under high NO conditions, but at NO mixing ratios <5 ppbv the production of $RO_2$ exceeds the destruction, highlighting (if this $\alpha$ value is correct) that there is a missing $RO_2$ sink at the

lower NO concentrations. Tan et al., (2019) also report a missing $RO_2$ sink under low NO conditions during PRD and suggested that autoxidation of $RO_2$ species could account for this missing sink and may also possibly act as the missing source of OH identified under the low NO conditions. An additional first order reaction that converts $RO_2$ to OH at a rate of 0.1 $s^{-1}$ brings the P:D(OH) and P:D($RO_2$) ratios close to 1 at all NO mixing ratios >0.3 ppbv, but at low NO mixing ratios (0.1 – 0.3 ppbv range) an even slower rate of conversion is required, highlighting, as one might expect, that the overall rate of $RO_2$

isomerisation is variable and likely depends on the specific $RO_2$ species present at a particular time or location. In the PRD study (Tan et al., 2019), the $HO_2$ budget was closed when $\alpha = 0.95$ was used suggesting that acyl peroxy radicals and those derived from long-chain alkanes and monoterpenes only made up a very small fraction of the total $RO_2$ concentration.

Although revealing, this type of experimental budget analysis coupled with the radical observations is unable to differentiate

between different $RO_2$ types and so assumptions have to be made on the fraction of the total $RO_2$ that propagate to $HO_2$. In the following section, a box model constrained to the latest MCM scheme (MCM3.3.1) is used to predict the radical concentrations. The MCM is a near explicit model and, as such, treats the production, propagation and destruction of each $RO_2$ species present discretely and so can provide an insight into the rate at which different $RO_2$ species convert to $HO_2$ or to other $RO_2$ species (or, indeed to OH) and the impact this propagation has on NO to $NO_2$ conversion and, hence, $O_3$ production.


### 3.4 MCM modelled radical predictions and comparison with observations

The time-series of the model-predicted radical concentrations and a breakdown of the modelled OH reactivity from the base MCM model are overlaid with the observations in figure 2. The average diurnal of the measured and modelled radical and k(OH) profiles are also provided in figure 5. In contrast to the experimental budget analysis, the model predicted OH is in

excellent agreement with the observed OH throughout the campaign. This same model over-estimates $HO_2$, however, particularly during the daytime, but also during the evening when a small secondary peak in $HO_2$ is predicted but not observed. An exception to this trend occurs between the $9^{th} – 12^{th}$ June when elevated levels of NO were measured at the site during the day and on these days, the agreement between the observed $HO_2$ and the model is better. The over-prediction of $HO_2$ primarily occurs under the lower NO conditions that were typically observed during the afternoon hours; the skill of the model to predict

the radical concentrations as a function of NO is discussed further below. The model under-estimates total $RO_2$ throughout the measurement period, although the level of disagreement (in absolute concentration) is most severe from the $15^{th} – 22^{nd}$ June when NO concentrations were at their lowest. During this period, the observed $RO_2$ concentrations were most elevated relative to other times during the campaign, however, the model does not predict a similar increase in $RO_2$ concentrations during this



period relative to other times in the campaign. OH reactivity is under-estimated by the model, on average by ~10 s$^{-1}$. However,
between the 15$^{th}$ – 22$^{nd}$ June the average missing OH reactivity increases to ~ 13 s$^{-1}$. The model underestimation of OH
reactivity may, in part, contribute to the model under-estimation of RO$_2$ as the model is evidently underestimating the rate of
OH + VOC reactions which form RO$_2$. Including an additional reaction between OH and VOC to account for the missing
reactivity in the model and the impact this has on the modelled radical concentrations is investigated in section 3.6. Although
the model is able to capture the observed OH concentrations reasonably well, the model's failure to reproduce the observed
HO$_2$ and RO$_2$ (and in the base model, the OH reactivity) indicates the model is either missing or misrepresenting some key
reactions. Furthermore, the discrepancy between the model-predicted OH and OH budget analysis which highlighted a missing
OH source, suggests that the over-prediction of HO$_2$ is masking a missing OH source in the MCM model.

Qualitatively, the model over-estimation of HO$_2$ and under-estimation of RO$_2$ is consistent with the budget analysis which
identified a missing RO$_2$ production term and missing HO$_2$ destruction term which could be reconciled, in part, by slowing the
rate at which RO$_2$ propagate to HO$_2$. However, when the HO$_2$ measured to modelled ratio is binned against NO, differences
between the model and budget analyses become apparent (figure 6). The model over-predicts the observed HO$_2$ concentrations
at the lowest NO mixing ratios experienced (0.1 – 1 ppbv); this over-prediction can be reconciled (under the very lowest NO
conditions, <0.3 ppbv) when a loss of HO$_2$ to aerosols (calculated using Eq. 9, with an uptake coefficient of 0.2) is included in
the model. This demonstrates that a reduction in aerosol surface area has the potential to enhance HO$_2$ concentrations and
thereby increase photochemical ozone formation, but only under very low NO conditions. As there was little to no change in
the modelled HO$_2$ concentration upon inclusion of an heterogeneous loss term under the higher NO conditions, efforts to
reduce anthropogenic PM when NO is present (which is highly likely to be the case) would not be expected to lead to an
increase in HO$_2$ and, in turn, O$_3$ as was suggested from earlier modelling studies (Li et al., 2019). Between 1 – 5 ppbv NO,
the model is able to reproduce the observed HO$_2$ well (between the 9$^{th}$ – 12$^{th}$ June, the daytime NO concentrations fell within
this intermediate NO range, hence the good agreement between the model and observations on these days). In contrast with
the budget analysis, the model under-predicts HO$_2$ beyond 5 ppbv NO by up to a factor of 10 at the highest NO experienced
(see the 52 ppbv NO bin, figure 6, which includes NO mixing ratios up to 104 ppbv). The model under-predicts the observed
RO$_2$ over the whole NO range and, consistent with the RO$_2$ budget analysis, the under-prediction (in terms of %) is greatest at
the highest NO concentrations experienced during the morning hours. The model under-predicts the observed RO$_2$ by ~factor
of 70 in the highest NO mixing ratio bin-range whereas the destruction rate of RO$_2$ exceeded the production rate by a factor of
~10 in the budget analysis. This large under-prediction of RO$_2$ by the model under the highest NO concentrations is most likely
driving the differences noted between the P to D(HO$_2$) and the measured to modelled (HO$_2$) ratios at NO mixing ratios >5
ppbv. Previous radical studies made at urban sites which were influenced by a range of NO$_x$ concentrations have demonstrated
that the level of agreement between model predictions and the observations tends to vary with the level of NO: Models have a
tendency to under-predict the observed OH concentrations at NO mixing ratios below 1 ppbv (Lu et al., 2012; Lu et al., 2013;
Tan et al., 2017; Whalley et al., 2018) and RO$_2$ concentrations are increasingly under-predicted as NO concentrations rise (Tan
et al., 2017; Whalley et al., 2018; Slater et al., 2020).



Cl atoms, formed from the photolysis of nitryl chloride (ClNO$_2$) have been shown to act as a source of RO$_2$ (Riedel et al.,
2014; Bannan et al., 2015; Tan et al., 2017) and have also been investigated here to see if Cl chemistry can resolve the modelled
RO$_2$ under-prediction under the elevated NO concentrations which were typically observed during the mornings. ClNO$_2$ was
measured for part of the campaign (Zhou et al., 2018) and reached up to 1.44 ppbv during the night on the 12$^{th}$ – 13$^{th}$ June.
The Cl atom concentration, calculated from the concentration of ClNO$_2$, its photolysis rate to yield Cl (determined from the
observed actinic flux and published absorption cross section of ClNO$_2$) and the VOC loading, exceeded 4 x 10$^4$ atoms cm$^{-3}$
during the morning of the 13$^{th}$ June and exceeded 1 x 10$^4$ atoms cm$^{-3}$ on several other mornings (figure 7). During these times,
the modelled RO$_2$ concentrations increased, relative to the concentration in the base model, by up to 2.5 x 10$^8$ molecule cm$^{-3}$
which represents close to a 100% increase in the modelled RO$_2$ at these times. On several mornings (4$^{th}$, 5$^{th}$, 7$^{th}$ and 13$^{th}$ June)
this increase in RO$_2$ brought the model and measured RO$_2$ into close agreement. The production rate of RO$_2$ from Cl-initiated
VOC oxidation on these mornings would serve to enhance P(RO$_x$) to the rate of D(RO$_x$). However, on several nights, only low
concentrations of ClNO$_2$ were measured and only very low concentrations of Cl atoms were calculated to be present upon
sunrise and so, on these days, only modest enhancements (1 – 2 x 10$^7$ molecule cm$^{-3}$) in RO$_2$ concentrations were predicted by
the model and the large under-prediction in the RO$_2$ concentration on these mornings remained which may indicate that there
are other, overlooked, primary RO$_x$ sources in the experimental budget calculation besides missing Cl + VOC reactions. The
Cl atom concentration dropped off rapidly during the mornings with just ~100 atoms cm$^{-3}$ present by noon on most days and
so was unable to reconcile the magnitude of the RO$_2$ underestimation observed throughout the day.

### 3.5 Rate of Production and rate of Destruction analysis

A rate of production and rate of destruction analysis on model OH, HO$_2$ and RO$_2$ species (figure 8) highlights the main radical
sources and sinks in the base model. Consistent with earlier studies of radicals in urban locations, the photolysis of HONO is
the dominant primary source of radicals during the daytime, accounting for ~64 % of the primary radical production on average
during the day (05:00 – 19:30) throughout the campaign. The photolysis of O$_3$ and subsequent reaction of O($^1$D) with H$_2$O
vapour accounts for ~9 % of primary production during the day, whilst the photolysis of HCHO and other photo-labile VOCs
accounts for ~11 % of the radical production. Ozonolysis and nitrate radical (NO$_3$) reactions account for 9 % and 7 % of the
total radical production during the day, respectively. At night, both ozonolysis (~18 %) and nitrate radical reactions (~82 %)
are the source of radicals. The primary source of radicals from VOC+NO$_3$ reactions is ~ 1 ppbv hr$^{-1}$ during the night which is
sufficient to close the RO$_x$ experimental budget (figure 3).

Figure 9 highlights the rates of propagation in the model which transform OH to HO$_2$ and RO$_2$, RO$_2$ to HO$_2$ and HO$_2$ back to
OH. The rate of propagation is rapid and the secondary source of OH from HO$_2$ + NO is more than twice as large as the primary
production of OH from HONO photolysis. Approximately one third of the OH reacts with CO, O$_3$ or HCHO to form HO$_2$, just
over one third reacts with VOCs to form RO$_2$ and just under one third is lost by reaction with NO$_2$ forming nitric acid. In





contrast to London (Whalley et al., 2018), the majority of $RO_2$ formed during AIRPRO propagate to $HO_2$ and subsequently the majority of $HO_2$ propagates back to OH. From the model radical flux analysis, which takes into consideration the different types of $RO_2$ species present, a value of $\alpha = 0.87$ is derived (where $\alpha$ = the rate at which RO forms $RO_2$ or $RC(O)O_2$ divided by the rate of RO conversion to $HO_2$). Note, this fraction does not consider $RO_2$ and $RC(O)O_2$ termination reactions. In London, the model derived $\alpha$ was ~0.5 reflecting the presence of long-chain alkane-derived $RO_2$ species from diesel emissions and mono-terpenes. In Beijing, measurement of such long-chain VOC species could not be attempted, but these could have been present. A lumped mono-terpene signal was measured by PTR-ToF-MS and is included in the model, split equally between $\alpha$-pinene and limonene. The base model, on which the radical flux analysis was performed, under-predicts OH reactivity and so is likely missing $RO_2$ species from additional OH+VOC reactions which, depending on the $RO_2$ type may serve to reduce $\alpha$.

## 3.6 OH reactivity and missing OH reactivity

$NO_2$ was the single biggest contributor to the OH reactivity in Beijing with a campaign average contribution of 18.6% (figure 5). This is similar to the $NO_2$ contribution to OH reactivity observed in London (Whalley et al., 2016). NO contributed just 1.3% to the total reactivity in Beijing, compared to a 4.2% contribution in London (Whalley et al., 2016). In London, measured carbonyl species accounted for close to 20% of the OH reactivity budget, largely due to the high concentrations of HCHO (Whalley et al., 2016). In contrast, in Beijing, carbonyls accounted for just 3.8% of the measured k(OH). Alkenes and dialkenes were more prevalent in Beijing than in London and the dialkene group of VOCs (dominated by isoprene) accounted for 10.5% of the OH reactivity in Beijing compared to 1.8% in London (Whalley et al., 2016). Owing to the faster physical loss of secondary species in Beijing by ventilation compared to London (see section 2), the contribution that model-generated intermediate species made to the observed OH reactivity was 2.7% in Beijing vs 23.8% in London (Whalley et al., 2016). In contrast to Beijing, where approximately 30% of the measured reactivity remains unaccounted for, in London, the OH reactivity budget was largely closed (Whalley et al., 2016). In Beijing during the measurement period when the missing OH reactivity reached on average 13 $s^{-1}$ (15th – 22nd June), isoprene concentrations were elevated relative to earlier in the campaign (figure 1). Overall, much higher concentrations of isoprene were observed in Beijing than in London (Whalley et al., 2016), and so this may indicate that other biogenic species that were not measured, along with their oxidation products, may account for some of the missing OH reactivity in Beijing.

A series of model simulations have been performed where an additional OH to $RO_2$ reaction has been included to account for the missing reactivity at a given time (figure 10); the $RO_2$ formed has been varied to investigate the influence of different $RO_2$ types on the modelled radical concentrations. When OH converts to methyl peroxy radicals, the modelled $RO_2$ concentration increases by close to a factor of 2 on average, but just over a factor of 2 under-prediction of the observed $RO_2$ radicals remains. Unsurprisingly, it is the modelled fraction of $RO_2$ radicals that do not act as an $HO_2$ interference ($RO_2$-simple) that increase in this scenario and the model now only under-estimates this class of $RO_2$ species by a factor of 1.45, whilst $RO_2$-complex is still under-estimated by a factor of 6.2. When OH converts to $HOCH_2CH_2O_2$ (an $RO_2$ species that does act as an $HO_2$ interference,



formed from the reaction of OH with ethene), the modelled $RO_2$-complex fraction increases and the model under-estimation of $RO_2$-complex is reduced to a factor of 1.8 on average, with the largest under-predictions observed during the evening hours. In both these model simulations, the modelled over-prediction of $HO_2$ increases from the base model scenario as $CH_3O_2$ and $HOCH_2CH_2O_2$ both rapidly propagate to $HO_2$. The modelled OH concentration displays a modest decrease with the additional OH sink, however, this is largely compensated for by the increase in modelled $HO_2$ which enhances the secondary source of

OH from $HO_2$ + NO, and so, overall, the modelled OH concentration is largely buffered by the inclusion of missing OH reactivity in the form of additional methane (leading to $CH_3O_2$) or ethene (leading to $HOCH_2CH_2O_2$)

Model simulations (not shown) which include an additional source of $CH_3C(O)O_2$, for example, from additional $CH_3CHO$+OH reactions, do predict substantially less $HO_2$ (and can reconcile the observed $HO_2$ to with 25%), but modelled $RO_2$

concentrations do not increase as a large fraction of the acyl-$RO_2$ radicals react with $NO_2$ to form PAN and are, therefore, lost. These missing reactivity model simulations and measurement comparisons suggest that the missing $RO_2$ may be a species which, upon reaction with NO, converts from one $RO_2$ species to another and, therefore, compete with $RO_2$ to $HO_2$ propagation rather than a $RO_2$ radical which lead to $RO_2$ termination. This suggests that the overall lifetime of $RO_2$ radicals is longer than currently estimated and that multiple conversions of one $RO_2$ species to another may be occurring to sustain the high

concentrations observed. As identified in London, larger, more complex VOC species such as mono-terpenes or long-chain alkanes deriving from diesel emissions do undergo multiple $RO_2$ to $RO_2$ conversions in the presence of NO as the alkoxy radical formed preferentially undergoes isomerisation rather than an external H atom abstraction by $O_2$. If an additional reaction which converts OH to an $RO_2$ species formed during the oxidation of α-pinene, and which undergoes four reactions with NO before eventually forming $HO_2$, is added to the model at a rate sufficient to reconcile the missing OH reactivity, the model

predicts significantly more total $RO_2$ and now only modestly under-predicts the observed $RO_2$ concentrations (by a factor of 1.8). In this simulation, both the simple- and complex-$RO_2$ species are enhanced, as the first 3 generations of $RO_2$ species formed would be detected during the $RO_x$-mode in the $RO_x$-LIF instrument and, hence, contribute to $RO_2$-simple. The final $RO_2$ species formed, that does propagate to $HO_2$ via RO upon reaction with NO, would be detected during the $HO_x$-mode in the $RO_x$LIF instrument and, as such, contributes to the $RO_2$-complex fraction. In this scenario, the $HO_2$ concentration is now

only modestly over-estimated by a factor of 1.4. The $RO_x$LIF instrument relies on the conversion of $RO_2$ species to $HO_2$ (and ultimately to OH) for detection, so one might expect the instrument to be insensitive to $RO_2$ species that do not directly propagate to RO then to $HO_2$ upon reaction with NO. However, given the $RO_x$LIF flow tube conditions (NO concentration of $4 \times 10^{13}$ molecule $cm^{-3}$ and residence time of just under 1 s) $RO_2$ species that require several reactions with NO before $HO_2$ is produced should still be detected. These types of $RO_2$ species that require more than one reaction with NO before $HO_2$ forms

may be generated via the additional VOC+OH reactions identified as missing OH reactivity (as presented here). They may also be present due to a missing primary source of $RO_2$ such as decomposition of a complex PAN species, VOC photolysis, a Cl atom + VOC reaction or an alkene ozonolysis product. The experimental peroxy radical budget analysis highlighted that budget closure could only be achieved if α was reduced to 0.1, which suggests that the model breakdown of peroxy radical



species present (e.g. the fraction of acyl-RO$_2$, long- vs short-chained alkyl-RO$_2$ species) may be incomplete. In the scenario
where OH converts to an α-pinene-derived RO$_2$ species, consistent with the experimental budget analysis, the model under-
predicts the observed OH by a factor of 1.8 revealing that there is a missing source of OH under the low NO conditions in
Beijing that was previously masked by the model over-prediction of HO$_2$.

### 3.7 Impact on ozone production


Previous work, for example, by Tan et al (2017), suggested that the addition of a primary RO$_2$ source could help reconcile the
model under-prediction of RO$_2$. However, as demonstrated in section 3.6, the identity of the primary RO$_2$ is important and in
Beijing a complex RO$_2$ species that has a large enough carbon skeleton such that the RO radical formed upon reaction with
NO preferentially isomerises to another RO$_2$ (and undergoes multiple RO$_2$ to RO$_2$ conversions before *eventually* forming HO$_2$)
is needed to reconcile both the observed RO$_2$ and HO$_2$ concentrations. These types of RO$_2$ species may also preferentially
isomerise rather than undergo the bimolecular reactions with NO if NO concentrations are low enough. For example, laboratory
studies have shown that the monoterpenes, following an initial attack by ozone or OH, form highly oxidised RO$_2$ radicals
within a few seconds via repeated H-shift from C– H to an R–O–O bond and subsequent O$_2$ additions (Jokinen et al., 2014;
Ehn et al., 2014; Berndt et al., 2016). Recently, autoxidation has also been shown to occur during the oxidation of aromatic
VOCs too (Wang et al., 2017b). Autoxidation reactions may generate OH directly from RO$_2$ and, therefore, may also resolve
the missing OH source reported under low NO conditions (here and in the literature). These types of autoxidation reactions
lead to the generation of HOMs also which have been shown to condense and contribute to SOA (Mohr et al., 2019). Mass
spectrometric signals relating to these highly oxidised RO$_2$ species were observed during the AIRPRO campaign (Brean et al.,
2019; Mehra et al., 2020) suggesting that autoxidation was occurring at the Beijing site. Unimolecular H-atom shifts are
represented within the MCM3.3.1 for isoprene oxidation. Autoxidation reactions for other RO$_2$ radicals are currently not
included within the MCM3.3.1, although improved representation of RO$_2$ radical chemistry is a focus for the next generation
of explicit detailed chemical mechanisms (Jenkin et al., 2019).

The model measurement comparisons above suggest that our understanding of the rate at which the larger RO$_2$ species
propagate to HO$_2$ (or to OH directly) and the possible reactions they undergo (which have not undergone substantial laboratory
study) is far from complete and highlights that RO$_2$ chemistry warrants further study. One important finding, however, is that
the underestimation of the observed RO$_2$ may be caused by missing reactions that compete with the RO$_2$+NO reactions that
form HO$_2$. These competing reactions are effectively slowing the rate at which RO$_2$ species convert to HO$_2$, but if, as suggested
here, these reactions are RO$_2$+NO reactions that reform another RO$_2$ radical, they will still be relevant in terms of ozone
production. Under low NO conditions there is emerging evidence that unimolecular isomerisation reactions occur for a range
of RO$_2$ radicals (Jokinen et al., 2014; Ehn et al., 2014; Berndt et al., 2016; Wang et al., 2017b); these reactions will effectively





remove RO$_2$ radicals without conversion of NO to NO$_2$ and so also have implications for modelling in situ O$_3$ production, if models rely only on the rate of VOC oxidation when investigating O$_3$ production.

By approximating the rate of ozone production to the rate of NO$_2$ production from the reaction of NO with HO$_2$ and RO$_2$ radicals, urban radical measurements can be used to estimate local ozone formation (Kanaya et al., 2007; Ren et al., 2013; Brune et al., 2016; Tan et al., 2017; Whalley et al., 2018). Losses of NO$_2$ that do not yield O$_3$, for example through nitric acid and PAN formation, need to be estimated and then subtracted:

$$P(O_3) = \left(k_{HO_2+NO}[HO_2][NO] + k_{RO_2+NO}[RO_2][NO]\right) - \left(k_{OH+NO_2+M}[OH][NO_2][M] + k_{RO_2+NO_2+M}[RO_2][NO_2][M]\right)$$

(11)

Using this approach, recent studies where OH, HO$_2$ and RO$_2$ observations (via RO$_x$LIF) were made, demonstrated that models may under-predict ozone production at high NO due to an underestimation of the RO$_2$ radical concentrations at high NO concentrations (Tan et al., 2017; Whalley et al., 2018). Figure 11 displays the mean ozone production calculated from the radical observations (red line) as a function of NO and, consistent with the earlier ozone production calculations from the Wangdu (Tan et al., 2017) and London (Whalley et al., 2018) studies, the in situ ozone production calculated from the modelled OH and peroxy radicals (black line) is lower than from the observed radicals, most significantly at the higher NO concentrations. To accurately simulate ozone production and to understand how emission reduction policies may impact ozone levels, it is essential that the model accurately reflects the types of RO$_2$ species present and how fast they propagate to another RO$_2$ species, or to HO$_2$ or to OH.

**5 Conclusions**

Measurement and model comparisons of OH, HO$_2$, RO$_2$-complex, RO$_2$-simple and total RO$_2$ in Beijing have displayed varying levels of agreement as a function of NO$_x$. Under low NO conditions, consistent with previous studies in low NO$_x$ but high VOC environments, a missing OH source is evident. Radical budget analysis has demonstrated that this missing OH source could be resolved if unimolecular reactions of RO$_2$ radicals generate OH directly. Under the low NO conditions (< 1 ppbv), the MCM over-predicted HO$_2$, although this over-prediction could be resolved at very low NO mixing ratios (<0.3 ppbv) by including a heterogeneous loss term to aerosol surfaces. This highlights that a reduction in aerosol surface area has the potential to enhance HO$_2$ concentrations and thereby increase photochemical ozone formation, but only under very low NO conditions. The model under-predicted RO$_2$, most severely under high NO conditions (>1 ppbv). Although Cl atoms could increase the concentration of RO$_2$, this enhancement was limited to times when the Cl atom concentration was elevated and could not resolve the RO$_2$ under-prediction observed at all times. In the presence of NO, the model over-estimates the rate at which RO$_2$ propagates to HO$_2$ and we hypothesise that larger RO$_2$ species likely undergo multiple bimolecular reactions with NO,



followed by isomerisation of the RO radical to another $RO_2$ species, before a $HO_2$ radical forms. By this process, the lifetime and the concentration of total-$RO_2$ radicals is extended. The ozone production efficiency of large, complex VOCs from which these $RO_2$ species are formed may be greater than currently appreciated, and so further efforts to understand the rate at which the larger $RO_2$ species propagate to $HO_2$ (or to OH directly) and all the possible reactions they undergo, is necessary to accurately model ozone levels in urban centres such as Beijing and to fully understand how emission controls will impact

ozone.

**Data availability.** Data presented in this study are available from the author upon request (l.k.whalley@leeds.ac.uk).

**Author contributions.** LW, ES, RWM, CY and DH carried out the measurements; LW and ES developed the model and performed the calculations; JL, FS, JH, RD, MS, JH, AL, AM, SW, AB, TB, HC, BO, RJ, LC, LK, WB, TV, SK, SG, YS, WX, SY, LR, WA, CH, XW and PF provided logistical support and supporting data to constrain the model; LW prepared the

manuscript, with contributions from all the co-authors.

**Competing interests**. The authors declare that they have no conflict of interest.

**Acknowledgements** – We are grateful to the Natural Environment Research Council for funding via the Newton Fund Atmospheric Pollution and Human Health in Chinese Megacity Directed International Program (grant number NE/N006895/1) and the National Natural Science Foundation of China (Grant No.41571130031). Eloise Slater, Freya Squires and Archit Mehra

acknowledge NERC SPHERES PhD studentships. We would like to thank Likun Xue and co-authors for the providing the chlorine chemistry module used in the MCM. We acknowledge the support from Zifa Wang and Jie Li from the Institute of Applied Physics (IAP), Chinese Academy of Sciences for hosting the APHH-Beijing campaign. We thank Liangfang Wei, Hong Ren, Qiaorong Xie, Wanyu Zhao, Linjie Li, Ping Li, Shengjie Hou and Qingqing Wang from IAP, Kebin He and Xiaoting Cheng from Tsinghua University, and James Allan from the University of Manchester for providing logistic and

scientific support for the field campaigns. We would also like to thank other participants in the APHH field campaign.

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



**Table 1: Chemical reactions that were used in the experimental budget analysis for OH, HO₂, RO₂ and ROₓ. The rate coefficient at 298K are given in column 3; temperature dependent rate coefficients were used in the experimental budget analysis presented in section 3.3**

| No. | Reaction | Rate coefficient (298K) $cm^3$ molecule$^{-1}$ s$^{-1}$ |
|---|---|---|
| R1 | Alkene $+ O_3 \rightarrow OH, HO_2, RO_2 +$ products | Specific rate coefficients and radical yields for each alkene, taken from the MCM3.3.1 (Jenkin et al., 2015) |
| R2 | $NO + HO_2 \rightarrow OH + NO$ | $8.5 \times 10^{-12}$ |
| R3 | $O_3 + HO_2 \rightarrow OH + 2O_2$ | $2.0 \times 10^{-15}$ |
| R4 | $HCHO + OH + O_2 \rightarrow CO + HO_2 + H_2O$ | $8.4 \times 10^{-12}$ |
| R5 | $CO + OH + O_2 \rightarrow HO_2 + CO_2$ | $2.3 \times 10^{-13}$ |
| R6 | $RO_2 + NO \rightarrow RO + NO_2$ | $8.7 \times 10^{-12}$ |
| R7 | $HO_2 + NO \rightarrow OH + NO_2$ | $8.5 \times 10^{-12}$ |
| R8 | $HO_2 + O_3 \rightarrow OH + 2O_2$ | $2.0 \times 10^{-15}$ |
| R9 | $HO_2 + RO_2 \rightarrow ROOH + O_2$ | $2.3 \times 10^{-11}$ |
| R10 | $HO_2 + HO_2 \rightarrow H_2O_2 + O_2$ | $1.7 \times 10^{-12}$ |
|  | $HO_2 + HO_2 + H_2O \rightarrow H_2O_2 + H_2O + O_2$ | $6.4 \times 10^{-30}$ |
| R11 | $RO_2 + RO_2 \rightarrow$ products | $3.5 \times 10^{-13}$ |
| R12 | $OH + NO_2 \rightarrow HNO_3$ | $1.1 \times 10^{-11}$ |
| R13 | $OH + NO \rightarrow HONO$ | $7.5 \times 10^{-12}$ |




**Table 2: The species measured by DC-GC-FID and PTR-ToF-MS that have been used as constraints in the model**

| Instrument | Species | Reference |
|---|---|---|
| DC-GC_FID | $CH_4$, $C_2H_6$, $C_2H_4$, $C_3H_8$, $C_3H_6$, isobutane, butane, $C_2H_2$, trans-but-2-ene, but-1ene, Isobutene, cis-but-2-ene, 2-Methylbutane, pentane, 1,3-butadiene, trans-2-pentene, cis-2-pentene, 2-methylpetane, 3-methypetane, hexane, isoprene, heptane, Benzene, Toluene, o-xylene, $CH_3OH$, $CH_3OCH_3$, ethylbenzene, $CH_3CHO$, $C_2H_5OH$ | Hopkins et al. (2011) |
| PTR-ToF-MS | α-pinene, limonene, isopropylbenzene, propylbenzene, xylene, trimethylbenzene. | Huang et al. (2016) |





**Table 3: Different model scenarios that are discussed in section 3**

| Model Name | Description |
|---|---|
| **Base model** | As described in section 2.4 |
| **Base model-SA** | The base model with the inclusion of a first order loss process of $HO_2$ to aerosols calculated using Eq 9 with an uptake coefficient, $\gamma = 0.2$ |
| **Base model-Cl** | The base model with the inclusion of Cl atom chemistry, taken from (Xue et al., 2015) |
| **Missing k(OH) (OH to $CH_3O_2$)** | The base model with an additional reaction converting OH to $CH_3O_2$ at a rate equal to the missing reactivity |
| **Missing k(OH) (OH to $HOCH_2CH_2O_2$)** | The base model with an additional reaction converting OH to $HOCH_2CH_2O_2$ at a rate equal to the missing reactivity |
| **Missing k(OH) (OH to $CH_3C(O)O_2$)** | The base model with an additional reaction converting OH to $CH_3C(O)O_2$ at a rate equal to the missing reactivity |
| **Missing k(OH) (OH to C96O2)**<br><br>**C96O2 =**  | The base model with an additional reaction converting OH to C96O2 (which is an α-pinene derived $RO_2$ species) at a rate equal to the missing reactivity |






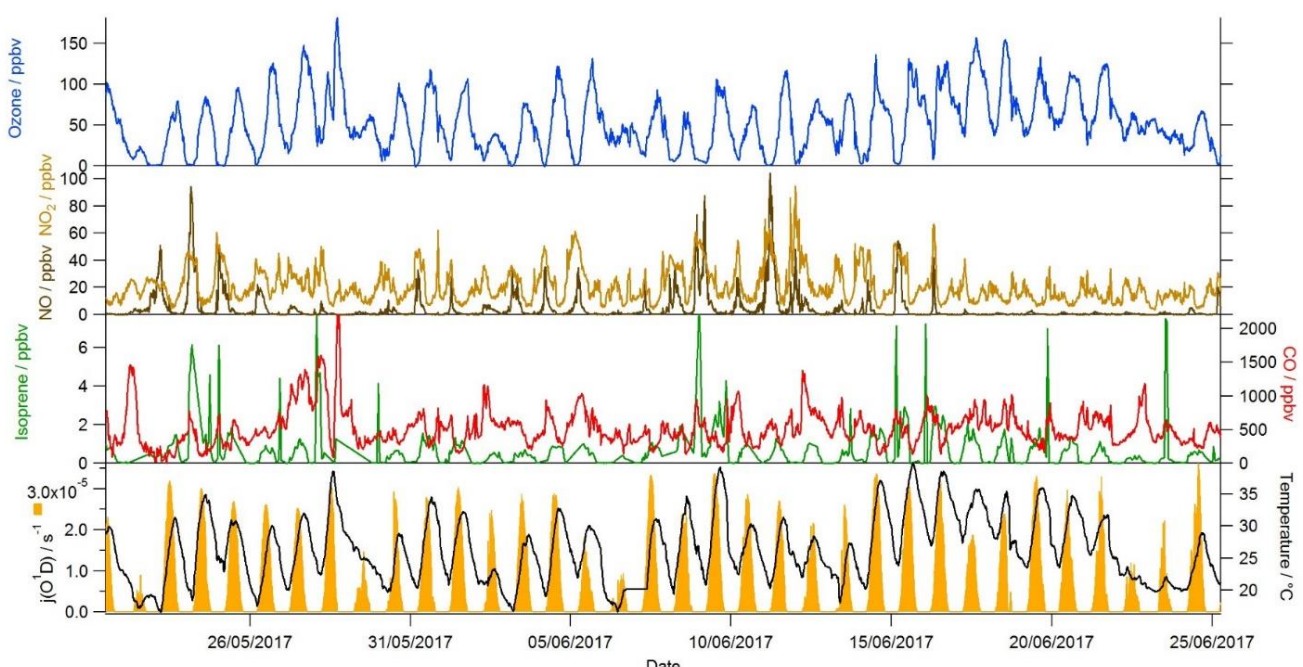

**Figure 1: Time-series of ozone, NO, NO₂ isoprene, CO, j(O¹D) and temperature during the campaign**

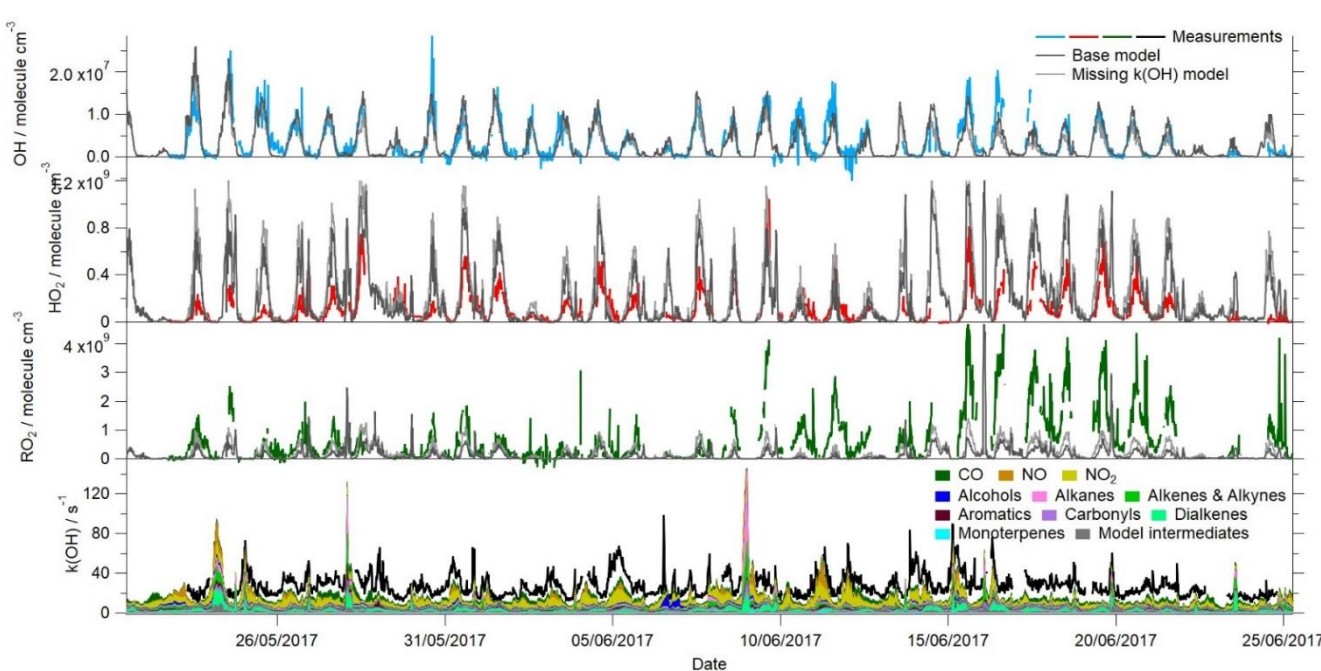


**Figure 2: Time-series of the measured and modelled OH, HO₂, total RO₂ and OH reactivity during the campaign**



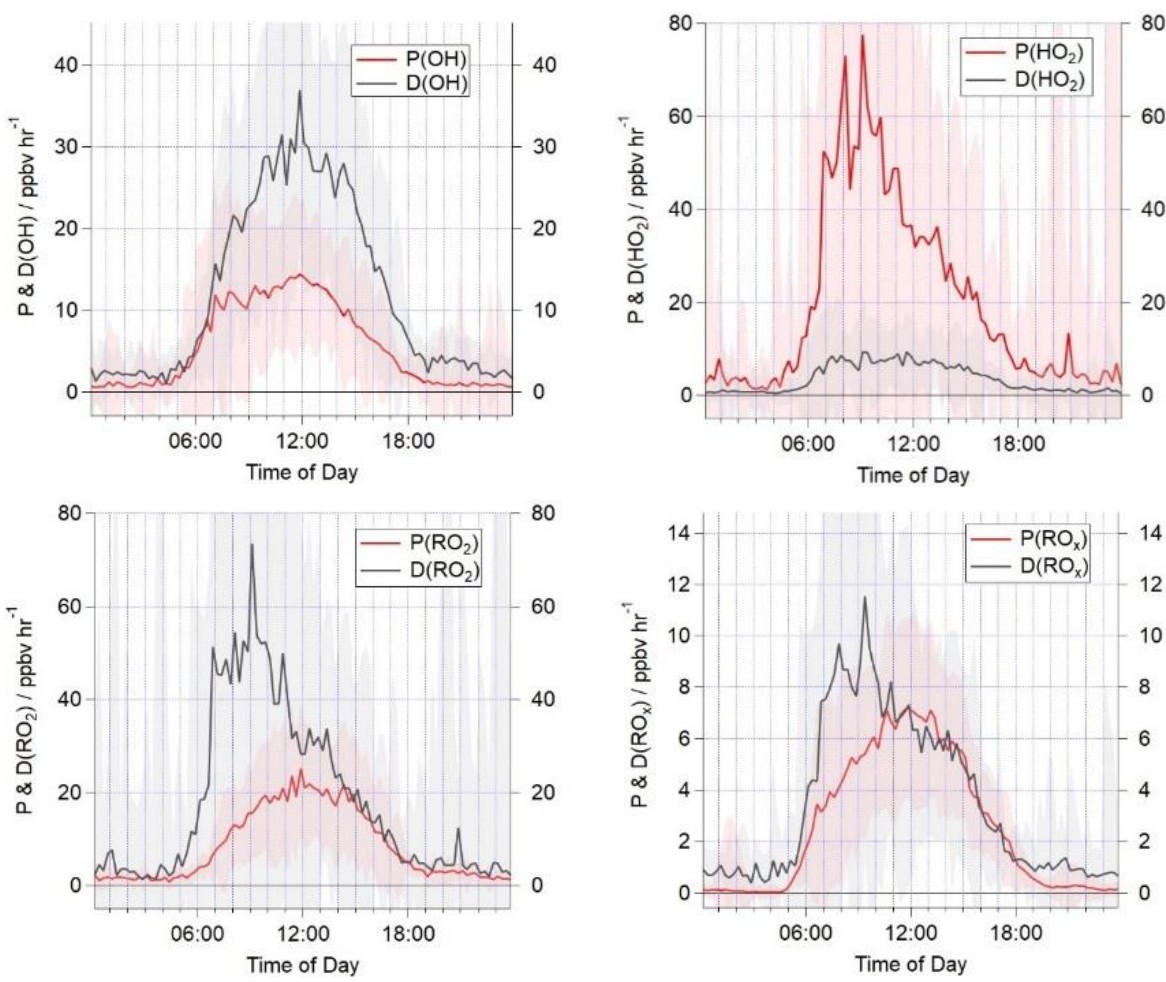

**Figure 3: Campaign median production and destruction rates for OH, HO₂, total RO₂ and ROₓ. The shaded areas represent the 1σ standard deviation of the data representing the variability from day to day**





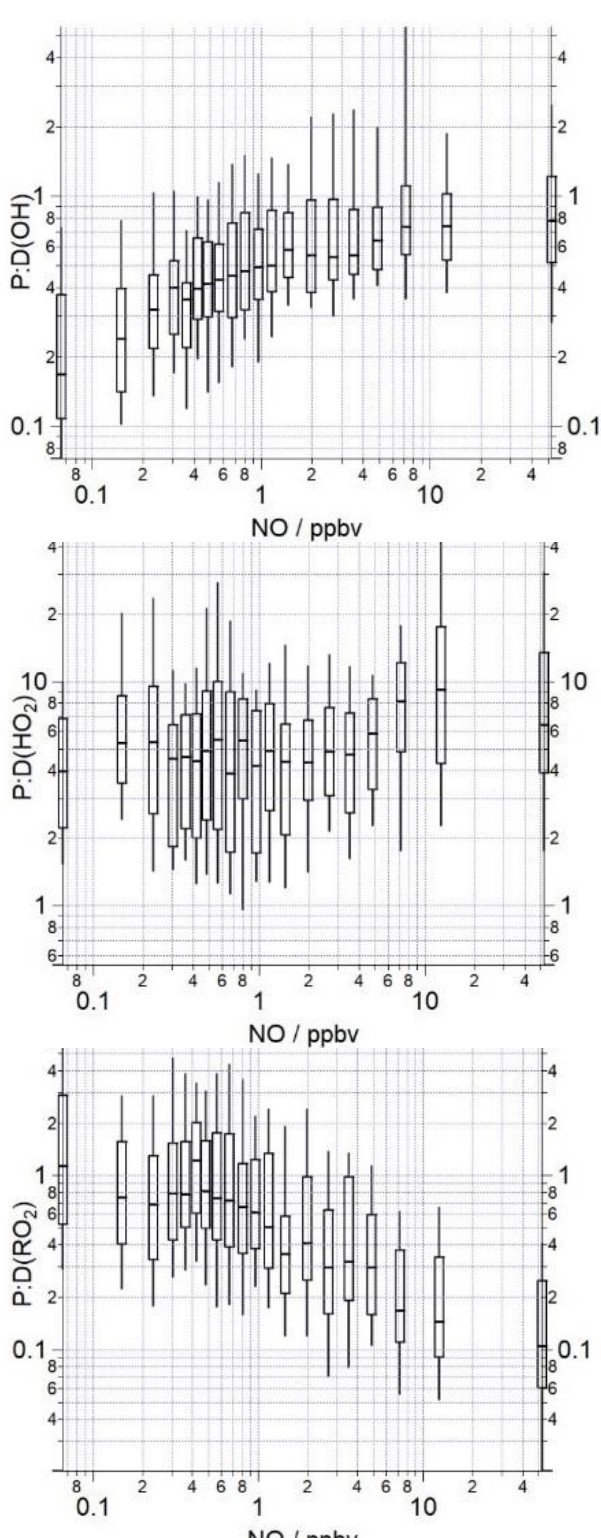

**Figure 4: The median ratio of the OH, HO₂ and total RO₂ production rates to destruction rates binned over the NO mixing ratio range encountered during the campaign on a logarithmic scale. The box and whiskers represent the 25th/75th and 5th/95th confidence intervals. The number of data points in each of the NO bins is ~80**

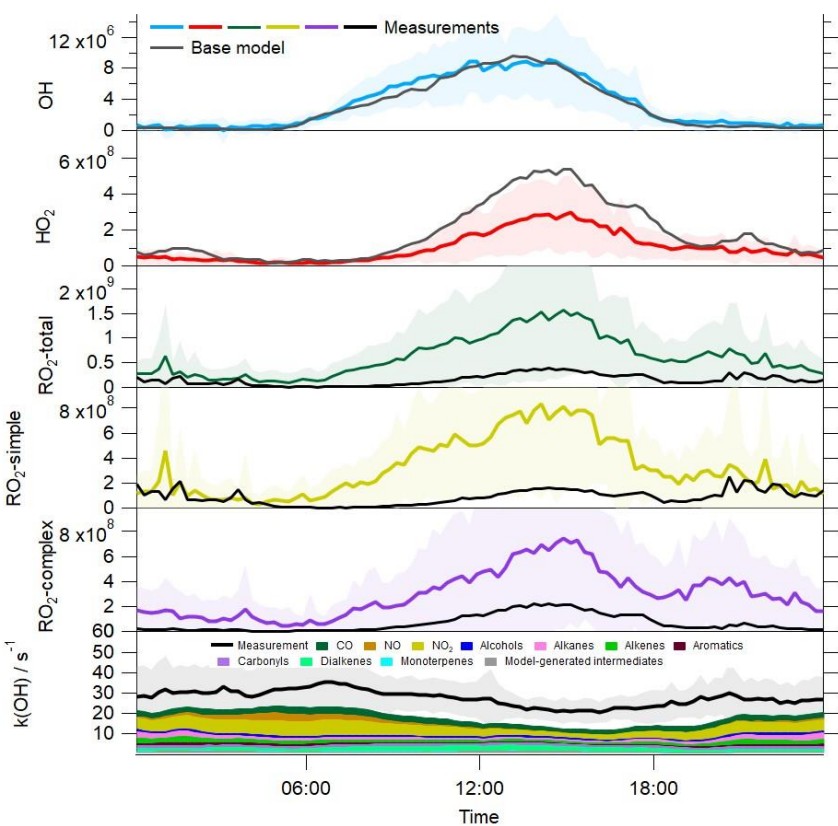

**Figure 5: Average profiles for the observed OH, HO₂, total RO₂, partially-speciated RO₂ (in molecule cm⁻³) and OH reactivity at 15 minute intervals over 24 hours. The error bars represent the 1 σ standard deviation of the measurements representing the variability in the measurements from day to day. The average diurnal profiles for OH, HO₂, total RO₂, partially speciated RO₂ and OH reactivity from the base model are overlaid**

990





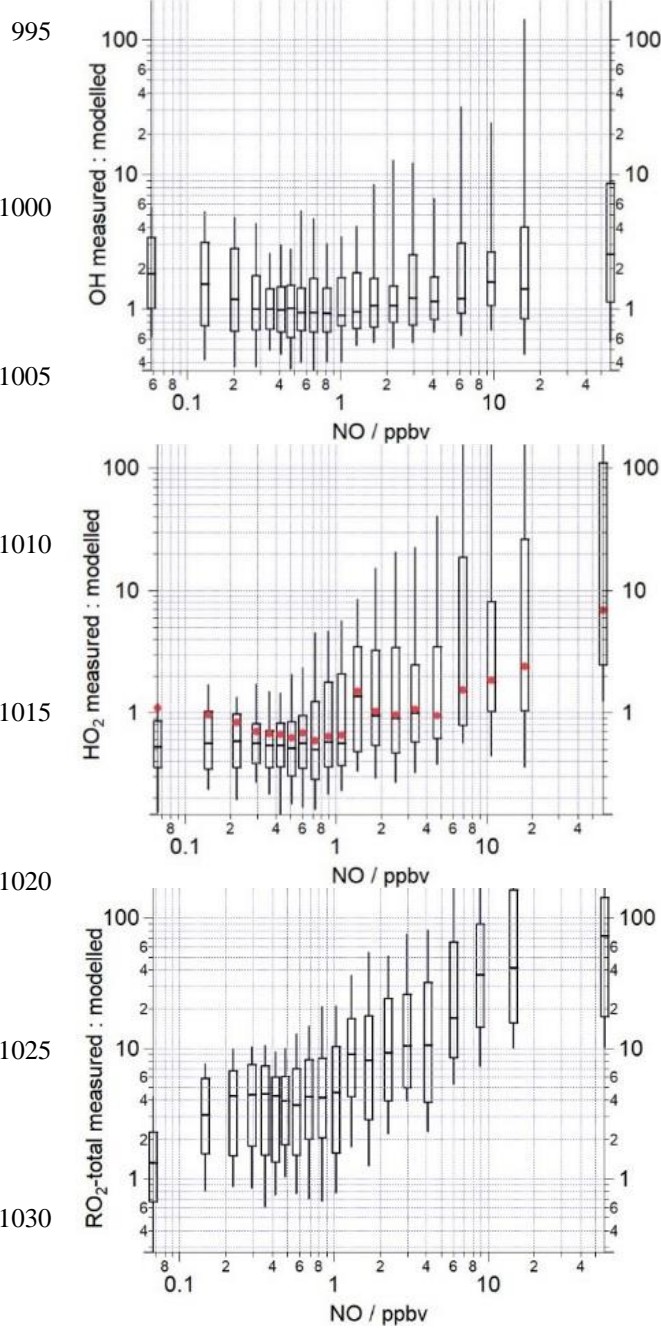

**Figure 6: The median ratio (-) of the measured to modelled OH, HO$_2$ and total RO$_2$ binned over the NO mixing ratio range encountered during the campaign on a logarithmic scale. The box and whiskers represent the 25th/75th and 5th/95th confidence intervals. The red circles in the middle panel display the measured to modelled HO$_2$ ratio when the model includes a heterogeneous loss of HO$_2$ to aerosols calculated using Eq. 9. The number of data points in each of the NO bins is ~80**



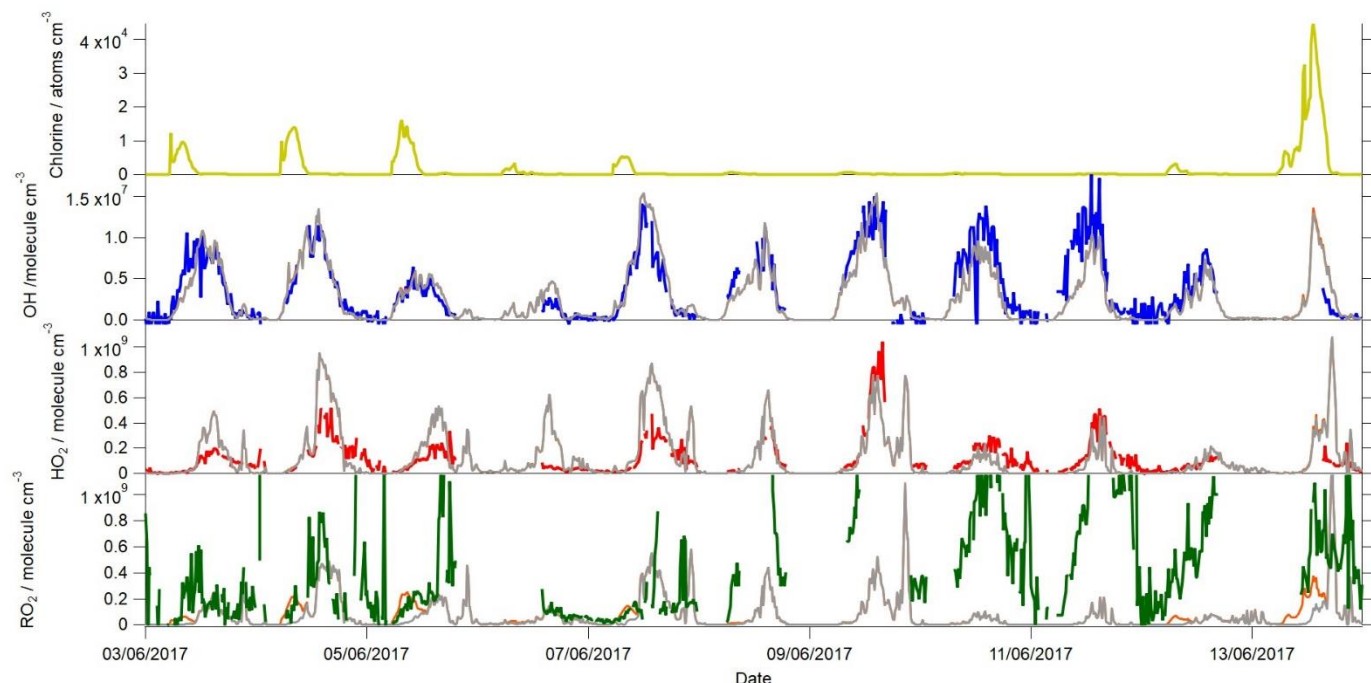

**Figure 7: Time-series of the measured and modelled OH, HO₂, total RO₂ during the campaign when ClNO₂ was also measured. The Cl atom concentration calculated to be present is shown in the top panel. The measured OH concentrations are represented by the blue line, HO₂ by the red line and total RO₂ by the green line. The base model scenario is shown in grey, whilst the base model**
**with Cl atom chemistry included (Xue et al., 2015) is shown in orange**



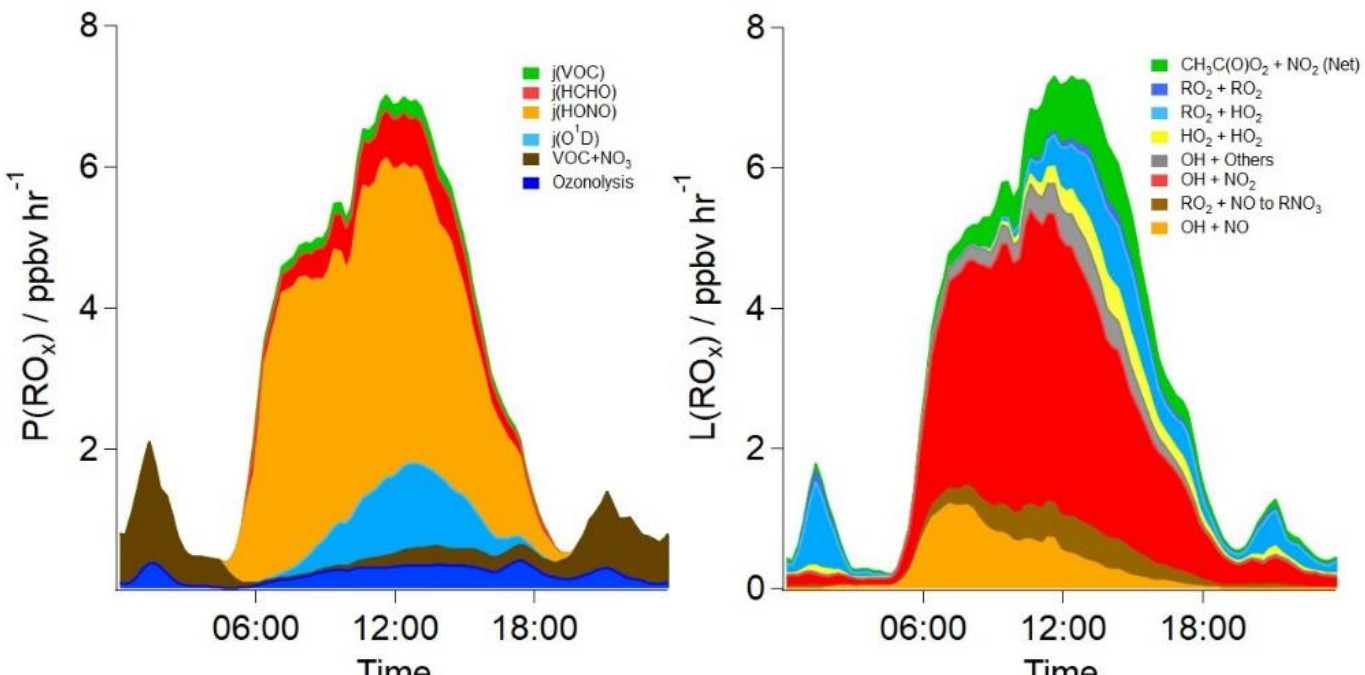

**Figure 8: The average diurnal rates of primary production and termination for RO$_x$ radicals in ppbv hr$^{-1}$ in the base model
scenario. CH$_3$C(O)O$_2$+ NO$_2$ (Net) represents the net rate (forward minus backward) for all RC(O)O$_2$+ NO$_2$ ↔ PAN species**





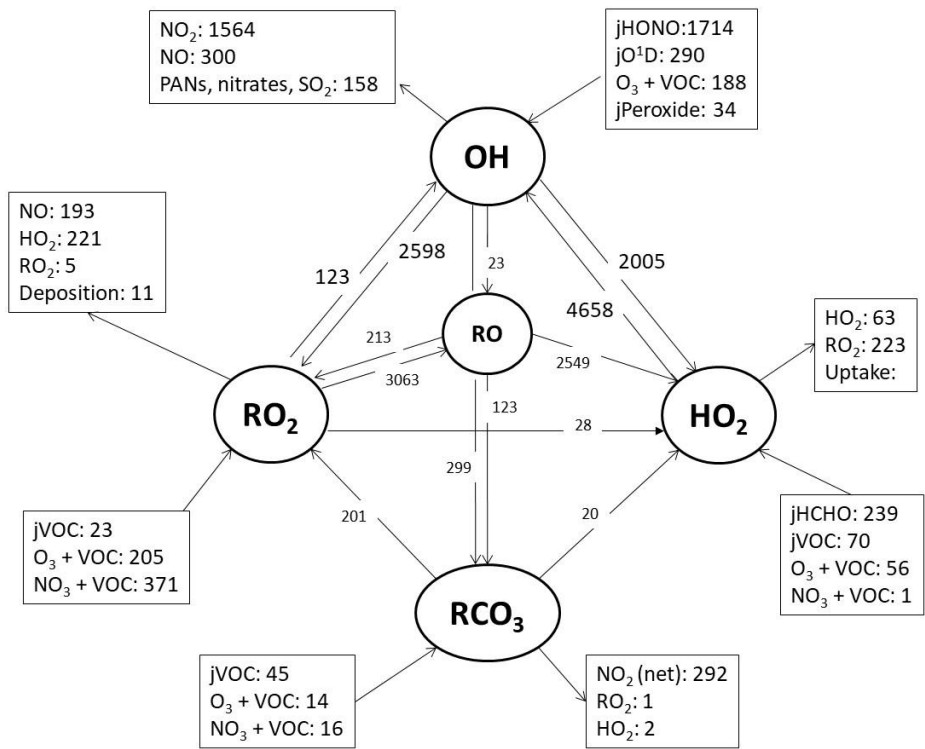

**Figure 9: A model reaction flux analysis, showing the mean rate of reaction for formation, propagation and termination of radicals (pptv hr$^{-1}$) (day and night) during the whole campaign**



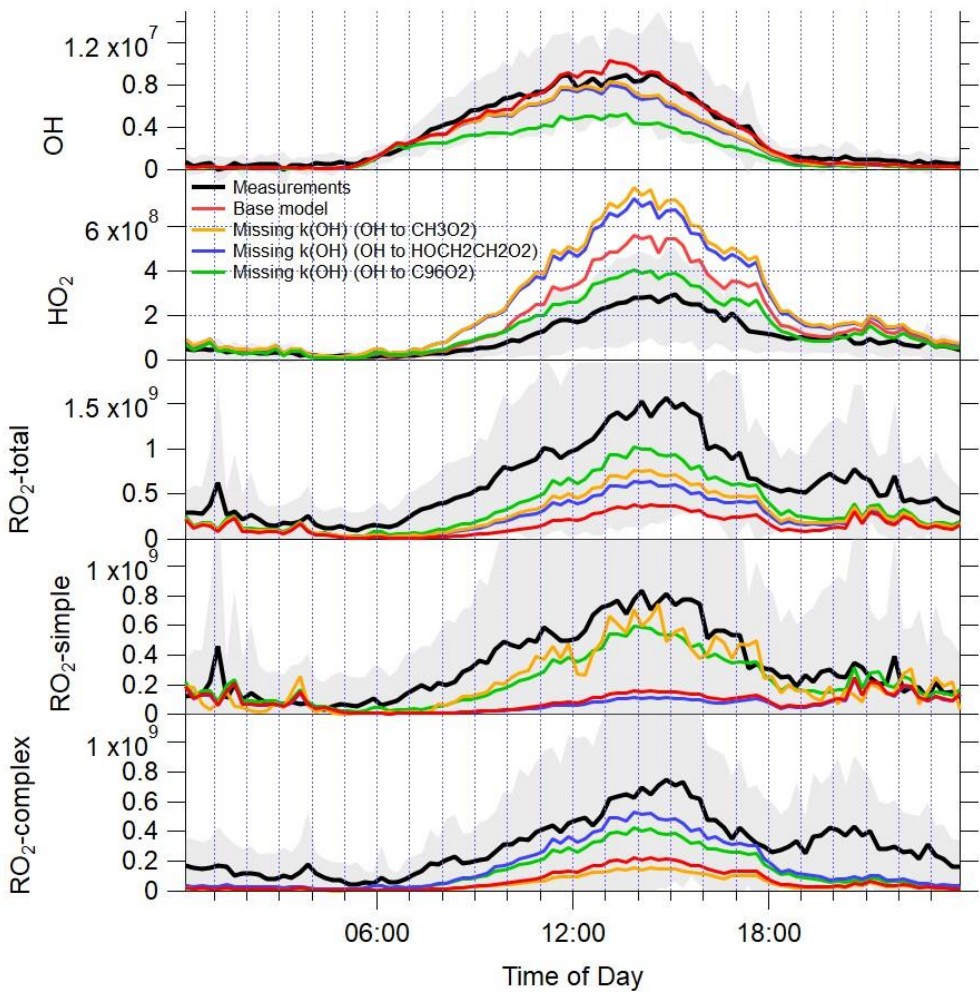

**Figure 10: Average diel profiles for the observed OH, HO₂, total RO₂, and partially-speciated RO₂ (black lines) at 15 minute intervals over 24 hours. The error bars represent the 1 σ standard deviation of the measurements. The average OH, HO₂, total RO₂ and partially speciated RO₂ model profiles when the missing reactivity observed at a given time is accounted for by different OH to RO₂ reactions are overlaid (yellow, blue and green lines); the base model predictions are in red. See text for details**





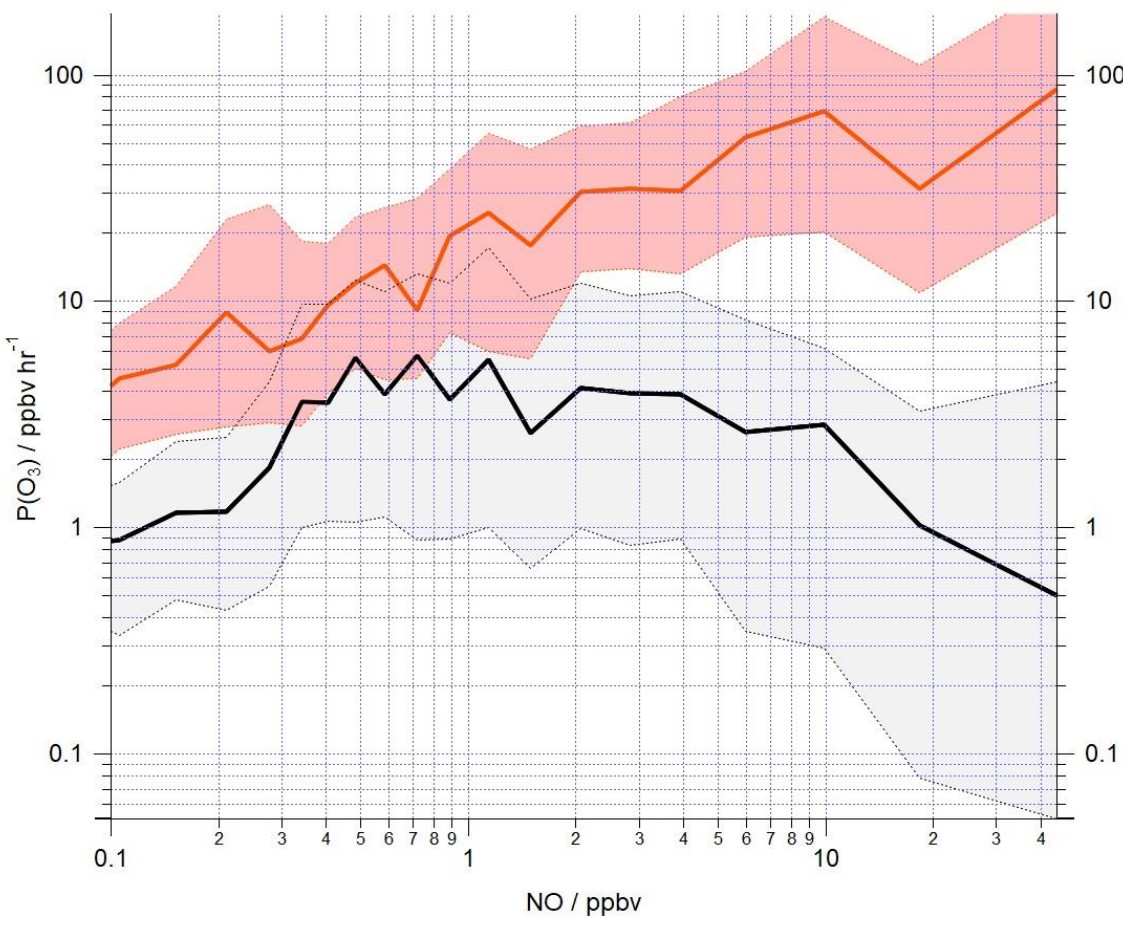

**Figure 11: Mean ozone production (ppbv hr$^{-1}$) calculated from observed (red line) and modelled (black line) $RO_x$ concentrations using Eq. (11) binned over the NO mixing ratio range encountered during the campaign on a logarithmic scale. The shading represents the 25$^{th}$ / 75$^{th}$ percentile confidence limits. The number of data points in each of the NO bins is ~80**