# Peer review of "Evaluating the sensitivity of radical chemistry and ozone formation to ambient VOCs and $NO_x$ in Beijing"

_Atmospheric Chemistry and Physics, 2020_

## Referee Comment (RC1) · Anonymous Referee #1 · 28 Oct 2020

This paper presents the measurements of OH, HO2, and RO2 radicals and OH re-activity in central Beijing in the summer of 2017 as part of the APHH campaign. It reportes the highest ever observed OH concentration of $2.8 \times 10^7$ cm-3 in urban area, even slightly higher than that reported in PRD in China by Lu et al. (2012).

Experimental budget analysis of OH, HO2, RO2, and ROx was performed in the similar way as Tan et al. (2019) did in PRD in 2014. Consistent with other studies in China, the authors found a missing OH source under low NO (<0.5 ppb) and high VOC condition. Besides, the authors found the opposite trends in HO2 budget and RO2 budget. The HO2 production rate exceeded the destruction rate by the similar rate as the RO2

destruction rate exceed production rate. The authors explained the opposite difference as the substantially slower than assumed net propagation rate of RO2 to HO2. If only 10% of the RO2 radicals propagate to HO2 upon reaction with NO, the HO2 and RO2 budget would be closed. The authors also performed a model simulation based on MCM 3.3.1, and found consistent results with the experimental budget analysis, except for the OH radical. The model simulated OH concentration very well due to a cancellation of missing OH source and sinks terms in its budget. The model underpredicted the kOH consistently across all NOx levels. To understand the model biases, the authors performed several sensitivity tests. The inclusion of heterogeneous loss of HO2 to aerosol surfaces and ClNO2 chemistry could not entirely explained the HO2 overestimation and RO2 underestimation, respectively. Several sensitivity tests were done to see the impact of missing OH reactivity on the modelled radical concentrations by assuming reactants convert OH to CH3O2, OHCH2CHO2O2, CH3(O)O2, and C96O2. The authors proposed that missing OH reactivity converted OH to a larger RO2 that undergo several reaction with NO, before eventually generating HO2, could improve the agreement between observation and simulation, and they used an $\alpha$-pinene-derived RO2 species (C96O2) as an example.

The results are of interest to the atmospheric chemistry community, enriching the ROx measurement in megacity, and the paper is worthy of publication. However, there are some critical issues and mistakes have to be addressed and corrected in advance before publication. Also, the paper could be shortened quite a bit and the writing could be more concisely and logically.

Specific comments:

1. Line 337, Alkyl nitrates are not formed from aldehydes + NO3.

2. According to the Fig.4, the RO2 neutral reaction rate (RO2+NO->RO2) has no dependence towards NO concentration since the P:D(HO2) showed no tendency towards lower NO. However, as the NO decreased, the competitive reaction of RO2 with HO2

or RO2 isomerization would become more and more important, and was even comparable to the rate between RO2 and NO. Thus, the multiple conversion of one RO2 to another should be reduced towards low NO.

3. The experimental configuration of RO2 convertor is missing.

4. In Line 573, the estimated NO concentration is the reactor is 4e13 cm-3. The reaction time scale of RO2+NO reaction is 0.003s. If such large flow was used in the reactor, the conversion to OH could be finished and the OH could further react with NO to form HONO. How do the author account for such conversion?

5. The RO2 and ROx budget is missing the part of Cl oxidation.

6. How sensitive of the experimental budget of HO2 and ROx radical towards the organic nitrate yield in the reaction of RO2 and NO? The organic nitrate yield varies from 0.01 to 0.5 among different RO2 species and it might have notable influence on the ROx and HO2 budget. Tan et al. (2019) not only set the yield to 0.05 but also performed the sensitivity tests by varying the yield from 5% to 20%, and notable influence was observed for their study although the bias was still within the experimental errors. Considering the large measured RO2 concentration, the yields might play significant role on this budget analysis in this study.

7. If it was the case as the author said, 90% of the measured RO2 would react with NO to produce another RO2, in which the majority of the RO2 was probably derived from long-chain alkanes, monoterpenes, and other like-VOCs, this part of RO2 should be detected in the RO2-complex. According to Fig 5, the RO2-complex only made up less than 50% of the total RO2. Besides, if the multiple bimolecular reaction of RO2 with NO made up such a proportion (90%), the ozone production would be inconceivably enhanced, but was not embodied in the observed O3 concentrations.

8. Line 563, Line 574-575, and Table 3, the author attributed the missing OH reactivity to additional reaction converting OH to C96O2, which is an $\alpha$-pinene derived RO2,

but C96O2 is formed in the $\alpha$-pinene reaction with O3 but NOT with OH. How do the authors justify this assumption? Some discussion to make such assumption is needed.

Technical comments:

1. Line 234, the last [RO2] should be out of the right bracket in Eq (6).

2. Line 360, 'production and destruction'.

3. There is no need for 2.4.1.

4. Line 513, $\alpha$ = 0.87 seems to be wrong or the description of $\alpha$ was confusing.

5. Conclusion should be section 4.

References

Lu, K. D., Rohrer, F., Holland, F., Fuchs, H., Bohn, B., Brauers, T., Chang, C. C., Häseler, R., Hu, M., Kita, K., Kondo, Y., Li, X., Lou, S. R., Nehr, S., Shao, M., Zeng, L. M., Wahner, A., Zhang, Y. H., and Hofzumahaus, A.: Observation and modelling of OH and HO$_2$ concentrations in the Pearl River Delta 2006: a missing OH source in a VOC rich atmosphere, Atmospheric Chemistry and Physics, 12, 1541-1569, 10.5194/acp-12-1541-2012, 2012.

Tan, Z., Lu, K., Hofzumahaus, A., Fuchs, H., Bohn, B., Holland, F., Liu, Y., Rohrer, F., Shao, M., Sun, K., Wu, Y., Zeng, L., Zhang, Y., Zou, Q., Kiendler-Scharr, A., Wahner, A., and Zhang, Y.: Experimental budgets of OH, HO2, and RO2 radicals and implications for ozone formation in the Pearl River Delta in China 2014, Atmospheric Chemistry and Physics, 19, 7129-7150, 10.5194/acp-19-7129-2019, 2019.
* * *

---

## Short Comment (SC1) · 28 Oct 2020

This paper presents some very interesting data and analysis from a study in Beijing using state-of-the-art measurements of OH, $HO_2$, and $RO_2$. Similar to a few other recent studies, the authors find that $RO_2$ concentrations and instantaneous ozone formation rates are both underestimated by 0-D models under high NOx conditions.

The authors define the instantaneous rate of ozone production using Equation 11:

$P(O_3) = (k_{HO2+NO}[HO_2][NO] + k_{RO2+NO}[RO_2][NO]) - (k_{OH+NO2+M}[OH][NO_2][M]+k_{RO2+NO2+M}[RO_2][NO_2][M])$

Similar definitions of $P(O_3)$ were used in Shirley et al. (2006), Sheehy et al. (2010), Dusanter et al. (2009), and Whalley et al. (2018), in contrast to the simpler earlier definitions which only included the first two terms on the right hand side of the equation, e.g., Kleinman et al. (1994), Thornton et al (2002), and Ren et al. (2003).

The last two terms are included to account for the fact that $O_3$ is not actually formed if an $NO_2$ molecule formed by the reaction of NO with $HO_2$ or $RO_2$ is then immediately removed by reaction with OH to form $HNO_3$ or with $RO_2$ to form a peroxy nitrate. The problem with this definition is that those two $NO_2$ removal reactions are just two of several Ox loss reactions, where $[Ox] = [O] + [O_3] + [NO_2] + [O(^1D)] + 2[NO_3] + 3[N_2O_5]$. For example, the reaction of $O(^1D)$ with $H_2O$ is just as much of an Ox loss mechanism as is the reaction of $NO_2$ with OH. Including only one Ox loss term in the definition of $P(O_3)$ is confusing and not quite accurate. It would be much simpler and more accurate to just define the rate of gross Ox production as

$P(O_X) = k_{HO2+NO}[HO_2][NO] + k_{RO2+NO}[RO_2][NO]$

and to separately define L(Ox), which would include the rates of the reactions $OH + NO_2$, $O(^1D) + H_2O$, $O_3 + HO_2$, etc. The net rate of peroxy nitrate ($RO_2NO_2$) formation or loss could also be included.

It is worth noting that truly defining the *instantaneous* formation rate of ozone (rather than Ox) necessitates accounting for variations in $j_{NO2}$, e.g. $P(O_3) = j_{NO2}[NO_2] - k[NO][O_3]$. The difficulty of evaluating this expression and its limited utility, especially on days with variable $j_{NO2}$ (due to clouds), underscore the advantage of considering Ox rather than $O_3$.

Please note the similar open comments made for Dusanter et al., (2009):

https://acp.copernicus.org/preprints/8/S5350/2008/acpd-8-S5350-2008.pdf

References

Dusanter, S., Vimal, D., Stevens, P. S., Volkamer, R., and Molina, L. T.: Measurements of OH and $HO_2$ concentrations during the MCMA-2006 field campaign Part 1: Deployment of the Indiana University laser-induced fluorescence instrument, Atmos. Chem. Phys., 9, 1665-1685, 2009.

Kleinman, L., Lee, Y. N., Springston, S. R., Nunnermacker, L., Zhou, X., Brown, R., Hallock, K., Klotz, P., Leahy, D., and Lee, J. H.: Ozone formation at a rural site in the southeastern United States, Journal of Geophysical Research: Atmospheres, 99, 3469-3482, 1994.

Ren, X. R., Harder, H., Martinez, M., Lesher, R. L., Oliger, A., Simpas, J. B., Brune, W. H., Schwab, J. J., Demerjian, K. L., He, Y., Zhou, X. L., and Gao, H. G.: OH and $HO_2$ chemistry in the urban atmosphere of New York City, Atmospheric Environment, 37, 3639-3651, 2003.

Sheehy, P. M., Volkamer, R., Molina, L. T., and Molina, M. J.: Oxidative capacity of the Mexico City atmosphere – Part 2: A ROx radical cycling perspective, Atmos. Chem. Phys., 10, 6993-7008, 10.5194/acp-10-6993-2010, 2010.

Shirley, T. R., Brune, W. H., Ren, X., Mao, J., Lesher, R., Cardenas, B., Volkamer, R., Molina, L. T., Molina, M. J., Lamb, B., Velasco, E., Jobson, T., and Alexander, M.: Atmospheric oxidation in the Mexico City Metropolitan Area (MCMA) during April 2003, Atmos. Chem. Phys., 6, 2753-2765, 2006.

Thornton, J. A., Wooldridge, P. J., Cohen, R. C., Martinez, M., Harder, H., Brune, W. H., Williams, E. J., Roberts, J. M., Fehsenfeld, F. C., Hall, S. R., Shetter, R. E., Wert, B. P., and Fried, A.: Ozone production rates as a function of $NO_x$ abundances and $HO_x$ production rates in the Nashville urban plume, Journal of Geophysical Research-Atmospheres, 107, 4146, 2002.

Whalley, L. K., Stone, D., Dunmore, R., Hamilton, J., Hopkins, J. R., Lee, J. D., Lewis, A. C., Williams, P., Kleffmann, J., Laufs, S., Woodward-Massey, R., and Heard, D. E.: Understanding in situ ozone production in the summertime through radical observations and modelling studies during the Clean air for London project (ClearfLo), Atmos. Chem. Phys., 18, 2547-2571, 10.5194/acp-18-2547-2018, 2018.

---

## Referee Comment (RC2) · Anonymous Referee #2 · 1 Nov 2020

This paper presents measurements of OH, HO2, and RO2 radical concentrations in addition to measurements of total OH reactivity in Beijing during the AIRPRO campaign in summer 2017. A radical budget analysis using the measured sources and sinks of these radicals revealed a potential missing source of OH during most of the campaign, although rates of OH production and destruction were in better balance under the higher NOx periods. The measured rates of HO2 production were found to be significantly greater than the rates of destruction, while the measured rates of destruction of RO2 radicals was found to be greater than the rates of production, especially under the higher NOx periods. These results suggest that the rate of conversion of RO2 to HO2 may be significantly slower than currently assumed.

[Figure]

The authors also present the results of several 0-D box models using the MCM 3.3.1 chemical mechanism. The model was able to reproduce the measured OH concentrations, but underestimated the measured total OH reactivity, suggesting that the agreement may be fortuitous. The model also overestimated the measured HO2 concentrations and underestimated the measured RO2 concentrations, consistent with the experimental radical budget suggesting that the model may be overestimating the rate of conversion of RO2 to HO2 under high NO conditions. The model was found to be in better agreement with the measurements if the missing reactivity was assumed to be composed of VOCs that produced a-pinene derived RO2 radicals that upon reaction with NO results in isomerization reactions that reform other RO2 species before eventually producing HO2 effectively reducing the rate of conversion of RO2 radicals to HO2. While this model scenario improved the model agreement with the measurements of HO2 and RO2, it significantly underestimates the measured OH concentrations, consistent with a missing OH source. However, the proposed RO2 isomerization reactions may lead to the production of OH radicals and contribute to the missing OH source. The significant underestimation of the observed RO2 concentrations implies that the model is significantly underestimating the observed rate of ozone production under high NOx conditions.

The measurements appear to be of high quality and include measurements of unknown interferences, which except for a few instances were found to be negligible. The measured radical concentrations are consistent with previous ROx measurements in several urban areas and is of interest to the atmospheric chemistry community. I recommend publication after the authors have addressed the following comments.

1) The analysis generally focuses on the campaign average and the measurements under higher NOx conditions, but there is little discussion regarding the measurements under lower NO conditions, and in particular the extended period at the end of the campaign where the measured RO2 concentrations were the highest. The scale used in Figure 2 makes it difficult to see, but the discrepancy between the measurements

and the model appears to be as significant as the discrepancies at higher NOx for this period. Unfortunately, this is not apparent from the information provided in Figure 6. It is not clear whether the additional VOC reactivity producing RO2 radicals that isomerize after reaction with NO to form additional RO2 would improve the model agreement for this period, as it is not clear whether reaction with NO still dominates the fate of peroxy radicals during this portion of the campaign. While the manuscript is already long, it would still benefit from a discussion of this aspect of their measurements.

2) Related to this, Berndt et al. (2018) report that RO2 + RO2 accretion reactions for a-pinene may be significant under low NOx conditions, and this type of accretion reaction may also be important for the peroxy radicals of other large VOCs. It's not clear whether these reactions could impact the modeled RO2 concentrations overall, but could be important during the low NOx period at the end of the campaign when the RO2 concentrations are high. Given that the authors are hypothesizing that isomerization of peroxy radicals of large VOCs produce additional peroxy radicals, the authors should comment on the potential impact of these reactions on the model results.

3) The authors should provide plots of some of the diurnal averaged constraints for their model (NO, NO2, O3, CO, isoprene, etc.) to allow comparisons with other urban measurements and to put the results shown in Figure 5 into context. Adding the diurnal average of the low NOx period at the end of the campaign would also assist in interpreting the radical measurements during this period. This information could go into a supplement.

4) The definition of alpha on page 17 line 513 appears to be an error as it is not consistent with the value and the definition described on page 8 line 246. This should be clarified.

Reference Berndt, T.; Mender, B.; Scholz, W.; Fischer, L.; Herrmann, H.; Kulmala, M.; Hansel, A., Accretion Product Formation from Ozonolysis and OH Radical Reaction of alpha-Pinene: Mechanistic Insight and the Influence of Isoprene and Ethylene.

Environ. Sci. Technol. 2018, 52 (19), 11069-11077.

---

## Author Comment (AC1) · 2 Dec 2020

**Referee 1**

This paper presents the measurements of OH, HO2, and RO2 radicals and OH reactivity in central Beijing in the summer of 2017 as part of the APHH campaign. It reportes the highest ever observed OH concentration of $2.8 \times 10^7$ cm-3 in urban area, even slightly higher than that reported in PRD in China by Lu et al. (2012).

Experimental budget analysis of OH, HO2, RO2, and ROx was performed in the similar way as Tan et al. (2019) did in PRD in 2014. Consistent with other studies in China, the authors found a missing OH source under low NO (<0.5 ppbv) and high VOC condition. Besides, the authors found the opposite trends in HO2 budget and RO2 budget. The HO2 production rate exceeded the destruction rate by the similar rate as the RO2 destruction rate exceed production rate. The authors explained the opposite difference as the substantially slower than assumed net propagation rate of RO2 to HO2. If only 10% of the RO2 radicals propagate to HO2 upon reaction with NO, the HO2 and RO2 budget would be closed. The authors also performed a model simulation based on MCM 3.3.1, and found consistent results with the experimental budget analysis, except for the OH radical. The model simulated OH concentration very well due to a cancellation of missing OH source and sinks terms in its budget. The model underpredicted the kOH consistently across all NOx levels. To understand the model biases, the authors performed several sensitivity tests. The inclusion of heterogeneous loss of HO2 to aerosol surfaces and ClNO2 chemistry could not entirely explained the HO2 overestimation and RO2 underestimation, respectively. Several sensitivity tests were done to see the impact of missing OH reactivity on the modelled radical concentrations by assuming reactants convert OH to CH3O2, OHCH2CHO2O2, CH3(O)O2, and C96O2. The authors proposed that missing OH reactivity converted OH to a larger RO2 that undergo several reaction with NO, before eventually generating HO2, could improve the agreement between observation and simulation, and they used an α-pinene-derived RO2 species (C96O2) as an example. The results are of interest to the atmospheric chemistry community, enriching the ROx measurement in megacity, and the paper is worthy of publication. However, there are some critical issues and mistakes have to be addressed and corrected in advance before publication. Also, the paper could be shortened quite a bit and the writing could be more concisely and logically.

**We thank referee 1 for their useful comments and have responded to each specific comment in bold below. The changes to the manuscript that we will make are in red.**

1. Line 337, Alkyl nitrates are not formed from aldehydes + NO3.

**This was a typo and should have been:**

**Alkyl nitrates, formed from isoprene + NO3 were also enhanced at these times at this site (Reeves et al., 2019).**

**This will be corrected in the revised manuscript**

2. According to the Fig.4, the RO2 neutral reaction rate (RO2+NO->RO2) has no dependence towards NO concentration since the P:D(HO2) showed no tendency towards lower NO. However, as the NO decreased, the competitive reaction of RO2 with HO2 or RO2 isomerization would become more and more important, and was even comparable to the rate between RO2 and NO. Thus, the multiple conversion of one RO2 to another should be reduced towards low NO.

**In response to a comment from the second reviewer, we have added model scenario 'Missing k(OH) (OH to C96O2)' to part of the radical measurement time-series, alongside the base model scenario to highlight that additional VOC reactivity which produces RO₂ radicals that isomerise after reaction**

with NO is able to increase the modelled total RO$_2$ concentration both under the lower NO conditions experienced between the 16$^{th}$ – 22$^{nd}$ June as well as on the higher NO days 9$^{th}$ – 12$^{th}$ June indicating that NO is still at sufficient concentrations to dominate the fate of RO$_2$ between the 16$^{th}$ – 22$^{nd}$ June, despite NO concentrations being lower.

However, as the referee states, in the afternoon low-NO chemistry (e.g. RO$_2$+HO$_2$ reactions) does play a greater role (30%), see Newland et al., (2020). Under these conditions, the reaction of RO$_2$ with NO and, therefore, the subsequent RO isomerisation, becomes less efficient and this trend is demonstrated when we overlay the RO$_2$ median measured to modelled (Missing k(OH) (OH to C96O2)) ratio vs NO on figure S3. We hypothesise that the production rate of HO$_2$ exceeds the destruction rate of HO$_2$ by a similar amount across the whole NO range encountered because we are neglecting both RO$_2$+NO reactions that lead to an RO radical that is able to undergo isomerisation reactions which would serve to reduce alpha most strongly under high NO conditions, whilst under low NO conditions we are neglecting RO$_2$ unimolecular reactions which may form OH directly rather than HO$_2$.

Newland, M. J., Bryant, D. J., Dunmore, R., Bannan, T., Acton, W. J., Langford, B., Hopkins, J., Squires, F. A., Dixon, W. J., Drysdale, W. S., Ivatt, P. D., Evans, M. J., Edwards, P., Whalley, L. K., Heard, D. E., Slater, E. J., Woodward-Massey, R., Ye, C., Mehra, A., Worrall, S. D., Bacak, A., Coe, H., Percival, C., Hewitt, C. N., Lee, J. D., Cui, T. Q., Surratt, J. D., Wang, X., Lewis, A. C., Rickard, A. R., and Hamilton, J.: Rainforest-like atmospheric chemistry in a polluted megacity, Atmospheric Chemistry and Physics Discussions, 35, 2020.

**Pg 14, line 440 onwards: The model under-estimates total RO$_2$ throughout the measurement period, although the level of disagreement (in absolute concentration) is most severe from the 16th – 22nd June when NO concentrations were at their lowest. During this period, the average NO mixing ratio was ~0.4 ppbv during the afternoon hours, whilst the average NO mixing ratio for the entirety of the campaign was ~0.75 ppbv during the afternoons (Fig S1 in SI). The average peak NO mixing ratio observed in the morning between 16th – 22nd June was just over 6 ppbv, whilst the average peak NO mixing ratio for the entirety of the campaign was close to 16 ppbv.**

**Pg 18, line 566 onwards: The modelled radical concentrations predicted from the 'Missing k(OH) (OH to C96O2)' scenario are overlaid with the radical observations and modelled radicals from the base model scenario in Fig S2, SI. The additional VOC reactivity which produces RO$_2$ radicals that isomerise after reaction with NO is able to increase the modelled total RO$_2$ concentration both under the lower NO conditions experienced between the 16$^{th}$ – 22$^{nd}$ June as well as on the higher NO days 9$^{th}$ – 12$^{th}$ June indicating that NO is still at sufficient concentrations to dominate the fate of RO$_2$ between the 16$^{th}$ – 22$^{nd}$ June, despite NO concentrations being lower. The median measured to modelled (Missing k(OH) (OH to C96O2)) ratio vs NO (Fig S3, SI) highlights that the inclusion of alkoxy isomerisation following RO$_2$ + NO reaction increases the modelled RO$_2$ across the entire NO range but, considering the log scale, has the biggest impact on the ratio (from the measured to modelled (base) ratio) at the highest NO concentration. Both the simple- and complex-RO$_2$ species are enhanced, as the first 3 generations of RO$_2$ species formed would be detected during the RO$_x$-mode in the RO$_x$-LIF instrument and, hence, contribute to RO$_2$-simple.**

**Supplementary Information**

[Figure]

**Figure S2: Time-series of the measured and modelled OH, HO₂, total RO₂ and OH reactivity from the 9th – 22nd June which encompasses high NO days (9th – 12th June) and low NO days (16th – 22nd June).**

[Figure]

[Figure]

**Figure S3: The median ratio (-) of the measured to modelled (base) OH, HO₂ and total RO₂ binned over the NO mixing ratio range encountered during the campaign on a logarithmic scale. The box and whiskers represent the 25th/75th and 5th/95th confidence intervals. The green circles display the measured to modelled OH, HO₂ and total RO₂ ratio when the model includes missing OH reactivity in the form of a single reaction which converts OH to C96O2. The number of data points in each of the NO bins is ~80**

**The median measured to modelled (Missing k(OH) (OH to C96O2)) ratio vs NO (green circles) is displayed in figure S3 alongside median measured to modelled (base) ratio. The inclusion of alkoxy isomerisation following RO₂ + NO reaction increases the modelled RO₂ concentration across the entire NO range but, considering the log scale, has the biggest impact on the ratio (from the measured to modelled (base) ratio) at the highest NO concentration. The HO₂ median measured to modelled (Missing k(OH) (OH to C96O2)) ratio vs NO in the middle panel increases from the measured to modelled (base) ratio at NO mixing ratios <1 ppbv, indicating improved agreement. At higher NO mixing ratios, where the base model begins to underpredict HO₂, due to the large under-prediction in RO₂, this under-prediction is reduced in the missing k(OH) (OH to C96O2) scenario owing to the increase in modelled RO₂.**

**The HO₂ median measured to modelled (Missing k(OH) (OH to C96O2)) ratio vs NO in the middle panel increases from the measured to modelled (base) ratio at NO mixing ratios <1 ppbv, indicating improved agreement. At higher NO mixing ratios, where the base model begins to underpredict HO2, due to the large under-prediction in RO₂, this under-prediction is reduced in the missing k(OH) (OH to C96O2) scenario owing to the increase in modelled RO₂.**

**The OH median measured to modelled (Missing k(OH) (OH to C96O2)) ratio vs NO (top panel) highlights a missing OH source, the magnitude of which deceases as NO concentrations increase.**

3. The experimental configuration of RO2 convertor is missing.

**We have provided an experimental description of the RO$_x$LIF instrument on pg 6 and provide references to previous papers (Whalley et al., 2018 and Slater et al., 2020) where further details can be found. We will add details on the physical dimensions of the RO$_2$ convertor to the revised manuscript.**

**Pg6, line 176: In the RO$_x$LIF reactor, which is an 83 cm long, 6.4 cm internal diameter flow-tube, in HO$_x$-mode, a flow of CO (10% in N$_2$) was added just beneath the sampling inlet and this rapidly converted any ambient OH sampled to HO$_2$. Within the RO$_x$LIF FAGE cell, a continuous flow of NO (99.95%) titrated ambient HO$_2$, the converted OH and also a large % of RO$_2$-complex radicals (see below) to OH for detection. In RO$_x$-mode, a total-RO$_2$ + HO$_2$ + OH measurement was made by addition of a dilute flow of NO (500 ppmv in N$_2$) alongside the CO which promoted the conversion of all HO$_2$ and RO$_2$ radicals to OH; the OH formed was rapidly re-converted to HO2 by reaction with CO. Within the RO$_x$LIF FAGE cell, the HO$_2$ was titrated back to OH, by reaction with NO, for detection.**

4. In Line 573, the estimated NO concentration is the reactor is 4e13 cm-3. The reaction time scale of RO2+NO reaction is 0.003s. If such large flow was used in the reactor, the conversion to OH could be finished and the OH could further react with NO to form HONO. How do the author account for such conversion?

**Excess CO (CO:NO = 50) was added continuously to the ROxLIF reactor, so the dominant reaction of OH, once formed, was with CO to reform HO$_2$ rather than reaction with NO.**

5. The RO2 and ROx budget is missing the part of Cl oxidation.

**Nitryl chloride measurements were only made for part of the campaign period, so it is not possible to add the production of RO$_2$ radicals from Cl atoms to the campaign averages. We will add the following sentence to the revised manuscript to help the reader gauge the impact Cl oxidation rates can have on RO$_2$ production:**

**Pg16, line 488: The production rate of RO$_2$ from Cl-initiated VOC oxidation on these mornings would serve to enhance P(RO$_x$) by up to 2.1 ppbv hr$^{-1}$.**

6. How sensitive of the experimental budget of HO2 and ROx radical towards the organic nitrate yield in the reaction of RO2 and NO? The organic nitrate yield varies from 0.01 to 0.5 among different RO2 species and it might have notable influence on the ROx and HO2 budget. Tan et al. (2019) not only set the yield to 0.05 but also performed the sensitivity tests by varying the yield from 5% to 20%, and notable influence was observed for their study although the bias was still within the experimental errors. Considering the large measured RO2 concentration, the yields might play significant role on this budget analysis in this study.

**Increasing the alkyl nitrate yield will decrease the production rate of HO$_2$, and would lead to an improved agreement with the HO$_2$ destruction rate. However, an increased alkyl nitrate yield would serve to increase both the RO$_2$ and the total RO$_x$ destruction rates, enhancing the discrepancy between the known RO$_2$ and RO$_x$ production rates further. The MCM, which takes into account the different types of RO$_2$ present from the VOCs observed, predicts a mean alkyl nitrate yield of 6%, so we feel that our choice of a 5% yield in the budget analysis is appropriate.**

7. If it was the case as the author said, 90% of the measured RO2 would react with NO to produce another RO2, in which the majority of the RO2 was probably derived from long-chain alkanes,

monoterpenes, and other like-VOCs, this part of RO2 should be detected in the RO2-complex. According to Fig 5, the RO2-complex only made up less than 50% of the total RO2.

**As we state on Pg 18, lines 566 – 569: both the simple- and complex-RO$_2$ species are enhanced (by including an alkoxy isomerisation mechanism in the model), as the first 3 generations of RO$_2$ species formed would be detected during the RO$_x$-mode in the RO$_x$-LIF instrument and, hence, contribute to RO$_2$-simple. The final RO$_2$ species formed, that does propagate to HO$_2$ via RO upon reaction with NO, would be detected during the HO$_x$-mode in the RO$_x$LIF instrument and, as such, contributes to the RO$_2$-complex fraction.**

Besides, if the multiple bimolecular reaction of RO2 with NO made up such a proportion (90%), the ozone production would be inconceivably enhanced, but was not embodied in the observed O3 concentrations.

**It is unwise to compare the ozone production rate to the observed ozone which will be impacted by physical processes such as advection, ventilation and deposition. The comparison of the model predicted glyoxal revealed that during the morning hours rapid ventilation effectively removed glyoxal from the model box. We can expect that ozone would be removed at the same rate and so the high rate of ozone production calculated from the observed peroxy radicals may not be reflected in the ozone concentration observed. We will comment on the losses of O$_x$ in the revised manuscript. See response to Ezra Wood's comment.**

**Pg20, line 613 onwards: By approximating the rate of ozone production to the rate of NO$_2$ production from the reaction of NO with HO$_2$ and RO$_2$ radicals, urban radical measurements can be used to estimate chemical ozone formation (Kanaya et al., 2007; Ren et al., 2013; Brune et al., 2016; Tan et al., 2017; Whalley et al., 2018).**

$$P(O_x) = \left( k_{HO_2+NO}[HO_2][NO] + k_{RO_2+NO}[RO_2][NO] \right) \tag{11}$$

**Losses of O$_x$ (L(O$_x$)) include chemical losses such as the reaction of NO$_2$ with OH, net PAN formation, the fraction of O($^1$D) (formed by the photolysis of O$_3$) that react with H$_2$O and the reaction of O$_3$ with OH and HO$_2$. Physical loss processes, such as O$_3$ deposition and ventilation out of the model box (see section 2.4) will also contribute to L(O$_x$). Physical processes such as advection of O$_3$ into the model box would also need to be considered in the model to make a direct comparison to the observed O$_3$ concentrations.**

**Considering the chemical production of O$_x$ (E.11), recent studies where OH, HO$_2$ and RO$_2$ observations (via RO$_x$LIF) were made, demonstrated that models may under-predict ozone production at high NO due to an underestimation of the RO$_2$ radical concentrations at high NO concentrations (Tan et al., 2017; Whalley et al., 2018).**

8. Line 563, Line 574-575, and Table 3, the author attributed the missing OH reactivity to additional reaction converting OH to C96O2, which is an α-pinene derived RO2, but C96O2 is formed in the α-pinene reaction with O3 but NOT with OH. How do the authors justify this assumption? Some discussion to make such assumption is needed.

**This is true, our motivation for choosing C96O2 was to investigate the impact of RO isomerisation forming RO2 in the model and so picked the C96O2 peroxy radical as this species undergoes several**

**isomerisation steps following RO₂+NO reaction and is already included in the MCM. We will add the following footnote to Table 3 in the revised manuscript to clarify this:**

**[1] Note, C96O2 is an α-pinene derived RO₂ that forms during the ozone-initiated oxidation of α-pinene. The additional production of C96O2 peroxy radicals in this model scenario was used to investigate the impact of an RO isomerisation mechanism on the modelled radical concentrations.**

Technical comments:

1. Line 234, the last [RO2] should be out of the right bracket in Eq (6).

**This will be corrected**

2. Line 360, 'production and destruction'.

**This will be corrected**

3. There is no need for 2.4.1.

**This will be removed and incorporated into section 2.4**

4. Line 513, α = 0.87 seems to be wrong or the description of α was confusing.

**The definition for alpha on page 17 line 513 should be α = 1 minus the rate at which RO forms RO2 or RC(O)O2 divided by the rate of RO conversion to HO2.**

5. Conclusion should be section 4.

**This will be corrected**

---

## Author Comment (AC2) · 2 Dec 2020

**Referee 2**

This paper presents measurements of OH, HO2, and RO2 radical concentrations in addition to measurements of total OH reactivity in Beijing during the AIRPRO campaign in summer 2017. A radical budget analysis using the measured sources and sinks of these radicals revealed a potential missing source of OH during most of the campaign, although rates of OH production and destruction were in better balance under the higher NOx periods. The measured rates of HO2 production were found to be significantly greater than the rates of destruction, while the measured rates of destruction of RO2 radicals was found to be greater than the rates of production, especially under the higher NOx periods. These results suggest that the rate of conversion of RO2 to HO2 may be significantly slower than currently assumed. The authors also present the results of several 0-D box models using the MCM 3.3.1 chemical mechanism. The model was able to reproduce the measured OH concentrations, but underestimated the measured total OH reactivity, suggesting that the agreement may be fortuitous. The model also overestimated the measured HO2 concentrations and underestimated the measured RO2 concentrations, consistent with the experimental radical budget suggesting that the model may be overestimating the rate of conversion of RO2 to HO2 under high NO conditions. The model was found to be in better agreement with the measurements if the missing reactivity was assumed to be composed of VOCs that produced a-pinene derived RO2 radicals that upon reaction with NO results in isomerization reactions that reform other RO2 species before eventually producing HO2 effectively reducing the rate of conversion of RO2 radicals to HO2. While this model scenario improved the model agreement with the measurements of HO2 and RO2, it significantly underestimates the measured OH concentrations, consistent with a missing OH source. However, the proposed RO2 isomerization reactions may lead to the production of OH radicals and contribute to the missing OH source. The significant underestimation of the observed RO2 concentrations implies that the model is significantly underestimating the observed rate of ozone production under high NOx conditions. The measurements appear to be of high quality and include measurements of unknown interferences, which except for a few instances were found to be negligible. The measured radical concentrations are consistent with previous ROx measurements in several urban areas and is of interest to the atmospheric chemistry community. I recommend publication after the authors have addressed the following comments.

**We thank referee 2 for their useful comments and have responded to each specific comment in bold below. The changes to the manuscript that we will make are in red.**

1) The analysis generally focuses on the campaign average and the measurements under higher NOx conditions, but there is little discussion regarding the measurements under lower NO conditions, and in particular the extended period at the end of the campaign where the measured RO2 concentrations were the highest. The scale used in Figure 2 makes it difficult to see, but the discrepancy between the measurements and the model appears to be as significant as the discrepancies at higher NOx for this period. Unfortunately, this is not apparent from the information provided in Figure 6. It is not clear whether the additional VOC reactivity producing RO2 radicals that isomerize after reaction with NO to form additional RO2 would improve the model agreement for this period, as it is not clear whether reaction with NO still dominates the fate of peroxy radicals during this portion of the campaign. While the manuscript is already long, it would still benefit from a discussion of this aspect of their measurements.

**We will extend the discussion on the model measurement comparison under the low NOx periods by including the following discussion:**

**Pg 14, line 440 onwards:** The model under-estimates total RO₂ throughout the measurement period, although the level of disagreement (in terms of absolute concentration) is most severe from the 16th – 22nd June when NO concentrations were at their lowest. During this period, the average NO mixing ratio was ~0.4 ppbv during the afternoon hours, whilst the average NO mixing ratio for the entirety of the campaign was ~0.75 ppbv during the afternoons (Fig S1 in SI). The average peak NO mixing ratio observed in the morning (16th – 22nd June) was just over 6 ppbv, whilst the average peak NO mixing ratio for the entirety of the campaign was close to 16 ppbv.

**Pg 18, line 566 onwards:** The modelled radical concentrations predicted from the 'Missing k(OH) (OH to C96O2)' scenario are overlaid with the radical observations and modelled radicals from the base model scenario in Fig S2, SI. The additional VOC reactivity which produces RO₂ radicals that isomerise after reaction with NO is able to increase the modelled total RO₂ concentration both under the lower NO conditions experienced between the 16th – 22nd June as well as on the higher NO days 9th – 12th June indicating that NO is still at sufficient concentrations to dominate the fate of RO₂ between the 16th – 22nd June, despite NO concentrations being lower. The median measured to modelled (Missing k(OH) (OH to C96O2)) ratio vs NO (Fig S3, SI) highlights that the inclusion of alkoxy isomerisation following RO₂ + NO reaction increases the modelled RO₂ across the entire NO range but, considering the log scale, has the biggest impact on the ratio (from the measured to modelled (base) ratio) at the highest NO concentration. Both the simple- and complex-RO₂ species are enhanced, as the first 3 generations of RO₂ species formed would be detected during the ROₓ-mode in the ROₓ-LIF instrument and, hence, contribute to RO₂-simple.

**Supplementary Information**

[Figure]

**Figure S2:** Time-series of the measured and modelled OH, HO₂, total RO₂ and OH reactivity from the 9th – 22nd June which encompasses high NO days (9th – 12th June) and low NO days (16th – 22nd June).

[Figure]

**Figure S3: The median ratio (-) of the measured to modelled (base) OH, HO$_2$ and total RO$_2$ binned over the NO mixing ratio range encountered during the campaign on a logarithmic scale. The box and whiskers represent the 25th/75th and 5th/95th confidence intervals. The green circles display the measured to modelled OH, HO$_2$ and total RO$_2$ ratio when the model includes missing OH reactivity in the form of a single reaction which converts OH to C96O2. The number of data points in each of the NO bins is ~80**

The median measured to modelled (Missing k(OH) (OH to C96O2)) ratio vs NO (green circles) is displayed in figure S3 alongside median measured to modelled (base) ratio. The inclusion of alkoxy isomerisation following $RO_2$ + NO reaction increases the modelled $RO_2$ concentration across the entire NO range but, considering the log scale, has the biggest impact on the ratio (from the measured to modelled (base) ratio) at the highest NO concentration. The $HO_2$ median measured to modelled (Missing k(OH) (OH to C96O2)) ratio vs NO in the middle panel increases from the measured to modelled (base) ratio at NO mixing ratios <1 ppbv, indicating improved agreement. At higher NO mixing ratios, where the base model begins to underpredict $HO_2$, due to the large under-prediction in $RO_2$, this under-prediction is reduced in the missing k(OH) (OH to C96O2) scenario owing to the increase in modelled $RO_2$.

The $HO_2$ median measured to modelled (Missing k(OH) (OH to C96O2)) ratio vs NO in the middle panel increases from the  measured to modelled (base) ratio at NO mixing ratios <1 ppbv, indicating improved agreement. At higher NO mixing ratios, where the base model begins to underpredict HO2, due to the large under-prediction in $RO_2$, this under-prediction is reduced in the missing k(OH) (OH to C96O2) scenario owing to the increase in modelled $RO_2$.

The OH median measured to modelled (Missing k(OH) (OH to C96O2)) ratio vs NO (top panel) highlights a missing OH source, the magnitude of which deceases as NO concentrations increase.

2) Related to this, Berndt et al. (2018) report that RO2 + RO2 accretion reactions for a-pinene may be significant under low NOx conditions, and this type of accretion reaction may also be important for the peroxy radicals of other large VOCs. It's not clear whether these reactions could impact the modeled RO2 concentrations overall, but could be important during the low NOx period at the end of the campaign when the RO2 concentrations are high. Given that the authors are hypothesizing that isomerization of peroxy radicals of large VOCs produce additional peroxy radicals, the authors should comment on the potential impact of these reactions on the model results.

We have taken the rate of accretion from Berndt et al and the observed $RO_2$ and NO concentrations to assess if accretion reactions may be competitive under low NO conditions experienced. If we assume that all $RO_2$ species measured undergo accretion reactions with a rate coefficient of $9.7 \times 10^{-12}$ $cm^3$ $molecule^{-1}$ $s^{-1}$, and compare this to the production rate of RO radicals from the reaction of $RO_2$ with NO we find that under the low $NO_x$ period, the production rate of accretion products is comparable to the production rate of alkoxy radicals. If we use the faster rate coefficient of accretion of $79 \times 10^{-12}$ $cm^3$ $molecule^{-1}$ $s^{-1}$, the production rate of accretion products is ~8.5 times faster than the RO production rate during the low $NO_x$ period, although this should be viewed as an upper limit as the total $RO_2$ concentration measured will contain a contribution from small $RO_2$ radicals, such as $CH_3O_2$, for which the rate of accretion is negligible. Nevertheless, we expect the inclusion of accretion reactions in the MCM would serve to reduce the modelled $RO_2$ concentration under low $NO_x$ conditions as the reaction represents a $RO_x$ sink. This suggests that the missing $RO_2$ source may be even larger than reported here. Accretion reactions effectively remove $RO_2$ radicals without conversion of NO to $NO_2$ and so have implications for modelling in situ $O_3$ production, if models rely only on the rate of VOC oxidation when investigating $O_3$ production.

We will add the following discussion to the manuscript:

**Pg 19, line 602 onwards: In addition to missing unimolecular $RO_2$ reactions, the model may be missing other $RO_2$ reaction pathways, for example, $RO_2$ accretion reactions, as identified by Berndt et al (2018). Although it is difficult to fully assess how competitive these $RO_2+RO_2$ reactions may be compared to $RO_2+NO$ reactions from the total $RO_2$ observations made (the concentration of each individual $RO_2$ would be needed), the inclusion of accretion reactions in the MCM would serve to reduce the modelled $RO_2$ concentration under low $NO_x$ conditions as the reaction represents an overall $RO_x$ sink. This suggests that the missing $RO_2$ source identified may be even larger under the lower NO conditions.**

**Pg 20, line 609 onwards: Under low NO conditions there is emerging evidence that unimolecular isomerisation reactions occur for a range of $RO_2$ radicals (Jokinen et al., 2014; Ehn et al., 2014; Berndt et al., 2016; Wang et al., 2017b) as well as $RO_2$ accretion reactions (Berndt et al., 2018). These reactions will effectively remove $RO_2$ radicals without conversion of NO to $NO_2$ and so also have implications for modelling in situ $O_3$ production, if models rely only on the rate of VOC oxidation when investigating $O_3$ production.**

3) The authors should provide plots of some of the diurnal averaged constraints for their model (NO, NO2, O3, CO, isoprene, etc.) to allow comparisons with other urban measurements and to put the results shown in Figure 5 into context. Adding the diurnal average of the low NOx period at the end of the campaign would also assist in interpreting the radical measurements during this period. This information could go into a supplement.

**We will include the following figure in the SI to assist in the interpretation of the radical observations and for comparison with other urban measurements.**

[Figure]

**Figure S1: Average profiles for the observed O₃, NO, NO₂, isoprene, and CO at 15 minute intervals over 24 hours. The solid lines represent the campaign average whilst the dashed line is the average NO profile between 16ᵗʰ – 22ⁿᵈ June.**

4) The definition of alpha on page 17 line 513 appears to be an error as it is not consistent with the value and the definition described on page 8 line 246. This should be clarified.

**The definition for alpha on page 17 line 513 should be $\alpha$ = 1 minus the rate at which RO forms $RO_2$ or $RC(O)O_2$ divided by the rate of RO conversion to $HO_2$.**

**This will be changed in the revised manuscript.**

---

## Author Comment (AC3) · 2 Dec 2020

**Short comment by Ezra Wood**

This paper presents some very interesting data and analysis from a study in Beijing using state of-the-art measurements of OH, HO2, and RO2. Similar to a few other recent studies, the authors find that RO2 concentrations and instantaneous ozone formation rates are both underestimated by 0-D models under high NOx conditions. The authors define the instantaneous rate of ozone production using Equation 11:

$P$(O3) = ($k$HO2+NO[HO2][NO] + $k$RO2+NO[RO2][NO]) − ($k$OH+NO2+M[OH][NO2][M]+$k$RO2+NO2+M[RO2][NO2][M])

Similar definitions of P(O3) were used in Shirley et al. (2006), Sheehy et al. (2010), Dusanter et al. (2009), and Whalley et al. (2018), in contrast to the simpler earlier definitions which only included the first two terms on the right hand side of the equation, e.g., Kleinman et al. (1994), Thornton et al (2002), and Ren et al. (2003). The last two terms are included to account for the fact that O3 is not actually formed if an NO2 molecule formed by the reaction of NO with HO2 or RO2 is then immediately removed by reaction with OH to form HNO3 or with RO2 to form a peroxy nitrate. The problem with this definition is that those two NO2 removal reactions are just two of several Ox loss reactions, where [Ox] = [O] + [O3] + [NO2] + [O(1D)] + 2[NO3] + 3[N2O5]. For example, the reaction of O(1D) with H2O is just as much of an Ox loss mechanism as is the reaction of NO2 with OH. Including only one Ox loss term in the definition of P(O3) is confusing and not quite accurate. It would be much simpler and more accurate to just define the rate of gross Ox production as P(OX) = $k$HO2+NO[HO2][NO] + $k$RO2+NO[RO2][NO] and to separately define L(Ox), which would include the rates of the reactions OH + NO2, O(1D) + H2O, O3 + HO2, etc. The net rate of peroxy nitrate (RO2NO2) formation or loss could also be included. It is worth noting that truly defining the instantaneous formation rate of ozone (rather than Ox) necessitates accounting for variations in jNO2, e.g. P(O3) = jNO2[NO2] − k[NO][O3]. The difficulty of evaluating this expression and its limited utility, especially on days with variable jNO2 (due to clouds), underscore the advantage of considering Ox rather than O3.

Please note the similar open comments made for Dusanter et al., (2009): https://acp.copernicus.org/preprints/8/S5350/2008/acpd-8-S5350-2008.pdf References Dusanter, S., Vimal, D., Stevens, P. S., Volkamer, R., and Molina, L. T.: Measurements of OH and HO2 concentrations during the MCMA-2006 field campaign Part 1: Deployment of the Indiana University laser-induced fluorescence instrument, Atmos. Chem. Phys., 9, 1665-1685, 2009.

**We thank Dr. Wood for his useful comment and agree that it would be more accurate to compare the modelled and measured P(O3) rather than the incomplete the net P(O3) that is in the manuscript currently. The loss terms in the calculation represent only a small subtraction and do not significantly change the differences reported between net P(O3) calculated from the measured and modelled peroxy radical concentrations. We will replace Figure 11 with the following figure which shows the modelled and measured P(O3) against NO and make the following changes to the text:**

**Pg20, line 613 onwards: By approximating the rate of ozone production to the rate of NO$_2$ production from the reaction of NO with HO$_2$ and RO$_2$ radicals, urban radical measurements can be**

used to estimate chemical ozone formation (Kanaya et al., 2007; Ren et al., 2013; Brune et al., 2016; Tan et al., 2017; Whalley et al., 2018).

$$P(O_x) = \left(k_{HO_2+NO}[HO_2][NO] + k_{RO_2+NO}[RO_2][NO]\right) \qquad (11)$$

Losses of $O_x$ ($L(O_x)$) include chemical losses such as the reaction of $NO_2$ with OH, net PAN formation, the fraction of $O(^1D)$ (formed by the photolysis of $O_3$) that react with $H_2O$ and the reaction of $O_3$ with OH and $HO_2$. Physical loss processes, such as $O_3$ deposition and ventilation out of the model box (see section 2.4) will also contribute to $L(O_x)$. Physical processes such as advection of $O_3$ into the model box would also need to be considered in the model to make a direct comparison to the observed $O_3$ concentrations.

Considering the chemical production of $O_x$ (Eq.11), recent studies where OH, $HO_2$ and $RO_2$ observations (via $RO_x$LIF) were made, demonstrated that models may under-predict ozone production at high NO due to an underestimation of the $RO_2$ radical concentrations at high NO concentrations (Tan et al., 2017; Whalley et al., 2018).

[Figure]

Figure 11: Mean $O_x$ production (ppbv hr−1) calculated from observed (red line) and modelled (black line) $RO_x$ concentrations using Eq. (11) binned over the NO mixing ratio range encountered during the campaign on a logarithmic scale. The shading represents the 25th / 75th percentile confidence limits. The number of data points in each of the NO bins is ~80